# $i$MIND: Insightful Multi-subject Invariant Neural Decoding

**Zixiang Yin, Jiarui Li, Zhengming Ding**[*]
Department of Computer Science, Tulane University
{zyin, jli78, zding1}@tulane.edu
https://zachyin.com/imind

## Abstract

Decoding visual signals holds an appealing potential to unravel the complexities of cognition and perception. While recent reconstruction tasks leverage powerful generative models to produce high-fidelity images from neural recordings, they often pay limited attention to the underlying neural representations and rely heavily on pretrained priors. As a result, they provide little insight into how individual voxels encode and differentiate semantic content or how these representations vary across subjects. To mitigate this gap, we present an *i*nsightful **M**ulti-subject **I**nvariant **N**eural **D**ecoding ($i$MIND) model, which employs a novel dual-decoding framework–both biometric and semantic decoding–to offer neural interpretability in a data-driven manner and deepen our understanding of brain-based visual functionalities. Our $i$MIND model operates through three core steps: establishing a shared neural representation space across subjects using a ViT-based masked autoencoder, disentangling neural features into complementary subject-specific and object-specific components, and performing dual decoding to support both biometric and semantic classification tasks. Experimental results demonstrate that $i$MIND achieves state-of-the-art decoding performance with minimal scalability limitations. Furthermore, $i$MIND empirically generates voxel-object activation fingerprints that reveal object-specific neural patterns and enable investigation of subject-specific variations in attention to identical stimuli. These findings provide a foundation for more interpretable and generalizable subject-invariant neural decoding, advancing our understanding of the voxel semantic selectivity as well as the neural vision processing dynamics.

## 1 Introduction

Deep learning models have recently been adopted in neuroscience as powerful tools for modeling brain activity, especially in the study of vision and cognition [6, 9, 19, 26]. A central goal in cognitive neuroscience is to understand how the brain transforms sensory input into meaningful representations that support recognition, memory, decision-making, and attention. Unlike behavioral annotations, neural signals provide a direct readout of these internal processes, revealing perceptual [3], emotional [32], and attentional dynamics [27] that cannot be fully captured by explicit labeling. Among available neural modalities, functional Magnetic Resonance Imaging (fMRI) [29] has been especially influential for its ability to non-invasively measure distributed patterns of cortical activity, enabling the study of how complex visual concepts are represented across brain regions. As such, fMRI not only offers a promising supervisory signal for aligning neural activity with computational models, but also serves as a critical tool for probing the neural basis of cognition and vision [16].

[*]Corresponding author. This work is partially supported by NIH 2U19 AG055373-06A1 as well as the Harold L. and Heather E. Jurist Center of Excellence for Artificial Intelligence at Tulane University.

39th Conference on Neural Information Processing Systems (NeurIPS 2025).

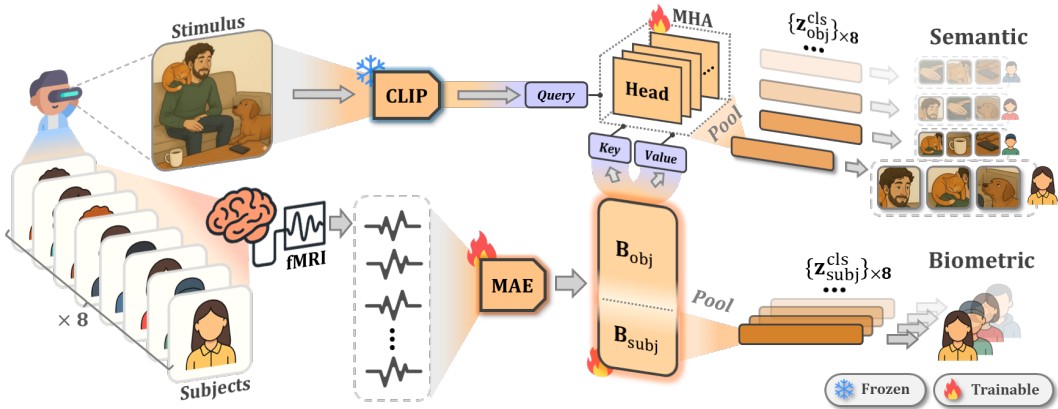

Figure 1: Overview of our framework. $i$MIND involves **biometric** decoding (identifying individuals) and **semantic** decoding (classifying perceived objects). fMRI voxel signals are first flattened and encoded by a pretrained Masked Autoencoder (MAE) to generate latent neural tokens, which are then passed through a learnable subject-object disentanglement block via orthogonal basis transformation $\mathbf{B} = (\mathbf{B}_{\text{subj}}, \mathbf{B}_{\text{obj}})$. The resulting biometric neural tokens are directly pooled for subject classification, while the semantic neural tokens act as *keys* and *values* in a cross-attention module with the CLIP latent features of the corresponding visual stimuli as *queries* to reflect the semantic contents captured by each subject. The fused representation is then pooled for multi-label object classification.

Most recent research has focused on reconstructing visual stimuli from brain activity by projecting neural representations into deep visual spaces and employing generative models such as GANs or diffusion models [5, 22, 35, 38, 41]. These approaches have yielded visually plausible, high-resolution images, suggesting that deep learning can serve as a bridge between brain activity and imagery. ***However, despite these remarkable achievements, we argue that reconstruction alone is fundamentally inadequate for understanding brain vision mechanism.*** Specifically, the following limitations exist:

- **Reconstruction relies heavily on pretrained generative priors**, which often dominate the decoding process and may introduce model-specific biases that obscure genuine neural content.

- **Brain recording may not encode all fine-grained details necessary for accurate reconstruction**– especially across subjects–rendering pixel-level reconstruction ill-posed and often misleading. This ultimately shifts the focus from reconstruction to generation.

Thus, we argue that reconstructing visual stimuli is neither a reliable nor interpretable strategy for decoding neural representations. A more effective alternative is to classify the subject's perceptual experience directly from fMRI data, enabling the identification of visual concepts embedded in neural responses [44]. This discriminative classification-based approach supports evaluation through standard metrics and allows researchers to disentangle both shared and individual-specific semantic components of brain activity–capabilities that reconstruction methods often fail to provide.

Technically, current neural decoding solutions follow two main strategies: single-subject models [5, 22, 24, 25, 35, 38], which suffer from data scarcity, overfitting, and poor scalability due to the high cost of fMRI collection; and multi-subject models [4, 36, 41, 44][2], which face substantial inter-subject variability from anatomical and functional differences. This can lead to entangled neural representations, where subject-specific and object-related signals become mixed [10], degrading decoding performance and interpretability. This raises a key research question:

> *"How can we develop a discriminative neural decoding framework that generalizes across individuals while preserving subject-specific semantic interpretations of visual stimuli?"*

To answer this, we present a novel approach to decoding brain activity using deep classification models, aiming to capture shared neural representations across individuals while preserving personalized interpretations shaped by diverse experiences and backgrounds. Toward this goal, we introduce the **i**nsightful **M**ulti-subject **I**nvariant **N**eural **D**ecoding ($i$**MIND**) model–a dual-decoding

---

[2]See Appendix A.1 for detailed related works.

framework (Figure 1) that supports both *biometric decoding* (identifying individuals) and *semantic decoding* (classifying perceived objects). By constructing voxel-object activation maps and examining subject-specific attentional patterns in response to complex, multi-object scenes, $i$MIND offers a principled path to disentangling individual and shared components of visual perception, ultimately advancing our understanding of brain-vision mechanisms. To sum up, our contributions are threefold:

- We introduce a multi-subject dual-decoding framework that disentangles neural signals into subject-specific and object-specific components, enabling scalable and subject-aware brain analysis with state-of-the-art semantic and biometric decoding performance.

- We develop a visual-neural interaction module that identifies object-voxel activation patterns in a data-driven manner, revealing how different subjects encode object-level semantics in the brain.

- We perform a comprehensive analysis of attention-based variability across subjects viewing the same visual stimuli, providing insights into individualized neural responses under time-constrained conditions.

## 2 Proposed Method

### 2.1 Problem Formulation

Consider a neural dataset containing brain activities of subjects $\mathcal{S}$ in response to visual stimuli drawn from an image dataset with $M$ samples $\mathcal{D} = \{(\mathbf{X}_i, \mathbf{y}_i)\}_{i=1}^M$, where each visual stimulus $\mathbf{X}_i$ is a three-channel image with a resolution of $H \times W$, paired with a **non-exclusive** ground-truth label $\mathbf{y}_i \in \mathbb{R}^C$ representing $C$ object categories. During neural recordings, all images $\mathbf{X}$ are viewed by a group of subjects $\mathcal{S}$. For each subject $s \in \mathcal{S}$ viewing the $i$-th image $\mathbf{X}_i$, an fMRI voxel response $\mathbf{V}_{i,s}$ is recorded, capturing neural activity specific to the subject-image pair. Note that the voxel signal $\mathbf{V}_{i,s}$ is flattened as an 1D vector with various lengths ($12{,}682 \sim 17{,}907$) across subjects due to biometric differences in neural structure. Following [5], we apply wrap-around padding to achieve a uniform voxel length $L$. Details on pre-processing procedures are provided in Appendix A.3.2.

### 2.2 Framework Overview

Our goal is to decouple arbitrary fMRI voxel signals into subject-invariant and subject-specific components from both semantic and biometric perspectives by developing a visual-neural model $\mathcal{M}$. Formally, this can be expressed as a mapping:

$$\mathcal{M} : \underbrace{\mathbb{R}^{H \times W \times 3}}_{\text{image}} \times \underbrace{\mathbb{R}^L}_{\text{voxels}} \to \underbrace{\mathbb{R}^C}_{\text{semantics}} \times \underbrace{\mathbb{R}^{|\mathcal{S}|}}_{\text{biometrics}} , \tag{1}$$

where *semantic decoding* seeks to extract object-related neural representations that are consistent across subjects, whereas *biometric decoding* focuses on capturing subject-specific neural patterns that are independent of the visual stimuli.

To address both tasks, we propose the **i**nsightful **M**ulti-subject **I**nvariant **N**eural **D**ecoding model, abbreviated as *i***MIND**. Building on the SC-MBM framework [5], our method constructs a $d$-dimensional shared latent neural space $\mathcal{F}$ across subjects to reduce the noise and redundancy inherent in fMRI signals [40]. Specifically, we employ a Vision Transformer (ViT)-based encoder $\mathcal{E} : \mathbb{R}^L \to \mathbb{R}^{N \times d}$ that projects each input voxel signal $\mathbf{V}$ into a set of $N$ neural feature tokens, denoted as $\mathbf{F} = \{\mathbf{f}_j\}_{j=1}^N$, where each $\mathbf{f}_j$ is a $d$-dimensional *neural feature token*. This yields $N$ neural representations per fMRI input, all embedded in the shared latent space $\mathcal{F}$. The encoder $\mathcal{E}(\cdot)$ is pretrained using a masked autoencoding objective in a self-supervised manner, emphasizing voxel-wise reconstruction.

Subsequently, we disentangle the learned features into object-specific and subject-specific components. The subject-specific features are used for subject classification (biometric decoding), while the object-specific features are aligned with frozen CLIP visual embeddings of the corresponding images via a cross-attention mechanism, enabling multi-label object classification (semantic decoding).

### 2.3 Subject-Object Disentanglement

The self-supervised pretraining reduces noise and redundancy in fMRI signals by encouraging the model to capture generalizable patterns. However, the resulting neural features $\mathcal{F}$ are not explicitly

tailored for downstream decoding tasks such as biometric identification or semantic classification. Because the reconstruction objective treats all latent information as equally relevant, it often leads to entangled representations–mixing subject-specific and object-specific components. This entanglement limits the interpretability of the learned features and hinders task-specific performance, especially when distinguishing between individual variability and shared visual semantics is essential.

Mathematically, this can be formalized by interpreting each neural feature token $\mathbf{f} = (f_1, f_2, \ldots, f_d) \in \mathbb{R}^d$ as the **coordinate representation** of a corresponding neural point $\mathbf{p}$ in the latent neural feature space $\mathcal{F}$. By default, this representation is expressed with respect to the standard basis $\mathbf{E} = \{\mathbf{e}_k \in \mathbb{R}^d\}|_{k=1}^d$ of $\mathbb{R}^d$, denoted as $[\mathbf{p}]_{\mathbf{E}}$:

$$[\mathbf{p}]_{\mathbf{E}} := \mathbf{f} = \sum_{k=1}^{d} f_k \mathbf{e}_k \in \mathbb{R}^d. \tag{2}$$

From this perspective, the entanglement of subject-specific and object-specific information within $\mathbf{f}$ can be attributed to the default use of $\mathbf{E}$ as the basis for spanning the neural feature space $\mathcal{F}$. As $\mathbf{E}$ is implicitly determined by the self-supervised task in the first training stage, this choice is beyond direct control, resulting in an inevitable blending of both subject and object information within the feature representation $\mathbf{f}$ for any neural point $\mathbf{p} \in \mathcal{F}$.

To enable effective downstream biometric and semantic decoding, we propose a solution from the perspective of feature disentanglement [8, 14, 39, 43]. We begin with assuming that the subject-specific and object-specific information within a neural point $\mathbf{p}$ are **linearly** entangled in the current neural feature representation $\mathbf{f}$ under the basis $\mathbf{E}$. While this may not fully characterize the complexities of neural dynamics, it serves as a simplified approximation to provide a meaningful step toward understanding the interplay between subject-specific and object-specific information. More crucially, this assumption applies at the **latent feature level**, not at the original fMRI signal level. At this level, the assumption is reasonable, as it aligns with the principles of deep classification tasks, where linear MLP classifiers rely on deep neural networks to transform inputs into linearly separable features for accurate classification. Based on this assumption, we resolve the linear entanglement by re-representing $\mathbf{p}$ with respect to a new basis $\mathbf{B}$. Specifically, we seek a **learnable** basis $\mathbf{B} = (\mathbf{B}_{\text{subj}}, \mathbf{B}_{\text{obj}})$ of $\mathbb{R}^d$ space that perfectly separates the subject-specific and object-specific features. Mathematically, this re-representation would allow $\mathbf{z}$, the representation of the same neural point $\mathbf{p}$ with respect to the new basis $\mathbf{B}$, to be distinctly split into subject-specific $\mathbf{z}_{\text{subj}}$ and object-specific $\mathbf{z}_{\text{obj}}$ components:

$$\mathbf{z} = [\mathbf{p}]_{\mathbf{B}} = \left([\mathbf{p}]_{\mathbf{B}_{\text{subj}}}, [\mathbf{p}]_{\mathbf{B}_{\text{obj}}}\right) = (\mathbf{z}_{\text{subj}}, \mathbf{z}_{\text{obj}}) \in \mathbb{R}^d. \tag{3}$$

According to the mathematical property of bases, the separation of $\mathbf{z}_{\text{subj}}$ and $\mathbf{z}_{\text{obj}}$ is guaranteed as long as $\mathbf{B}$ forms an **orthonormal** basis of $\mathbb{R}^d$, i.e., $\mathbf{B}\mathbf{B}^\top = \mathbf{I}_d$, where $\mathbf{I}_d$ is the identity matrix of rank $d$. With this orthonormality condition satisfied, the transformation from the original representation of $\mathbf{f}$ to the new representation $\mathbf{z}$ of the same neural point $\mathbf{p}$ can be derived as follows:

$$\mathbf{z} = [\mathbf{p}]_{\mathbf{B}} = \mathbf{B}^{-1}[\mathbf{p}]_{\mathbf{E}} = \mathbf{B}^\top[\mathbf{p}]_{\mathbf{E}} = \mathbf{B}^\top \mathbf{f} \in \mathbb{R}^d. \tag{4}$$

Combined with Eq. (3), we finally arrive at our subject-object disentanglement formulation:

$$\mathbf{z}_{\text{subj}} = \mathbf{B}_{\text{subj}}^\top \mathbf{f} \in \mathbb{R}^{d_{\text{subj}}} \quad \text{and} \quad \mathbf{z}_{\text{obj}} = \mathbf{B}_{\text{obj}}^\top \mathbf{f} \in \mathbb{R}^{d_{\text{obj}}}. \tag{5}$$

From a feature transformation perspective, we realize subject-object disentanglement by decomposing the original neural space $\mathcal{F}$ into two complementary (orthogonal) subspaces: the subject-specific neural subspace $\mathcal{F}_{\text{subj}}$ and the object-specific neural subspace $\mathcal{F}_{\text{obj}}$. This disentanglement is achieved by learning two orthonormal sets, $\mathbf{B}_{\text{subj}} \in \mathbb{R}^{d \times d_{\text{subj}}}$ and $\mathbf{B}_{\text{obj}} \in \mathbb{R}^{d \times d_{\text{obj}}}$, which span $\mathcal{F}_{\text{subj}}$ and $\mathcal{F}_{\text{obj}}$ respectively. In our framework, $d_{\text{obj}}$ is treated as a user-defined value, with $d_{\text{subj}} := d - d_{\text{obj}}$ to complete the basis. A formal and theoretic proof is provided in Appendix A.2.

The complementary relationship, $\mathcal{F} = \mathcal{F}_{\text{subj}} \oplus \mathcal{F}_{\text{obj}}$, guarantees a clear separation of subject and object information in the transformed neural representation $\mathbf{z} = (\mathbf{z}_{\text{subj}}, \mathbf{z}_{\text{obj}}) \in \mathbb{R}^d$ for each neural point $\mathbf{p} \in \mathcal{F}$, establishing a foundation for the subsequent biometric and semantic decoding tasks.

## 2.4 Biometric & Semantic Decoding

In this section, we describe our approach to decoding fMRI signals biometrically and semantically. Note that an fMRI signal $\mathbf{V}$ is represented by $\mathbf{F}$ as a set of $N$ neural tokens $\{\mathbf{f}_j\}_{j=1}^N$ within the

latent neural space $\mathcal{F}$. Based on Eq. (5), the subject-specific feature map $\mathbf{Z}_{\mathrm{subj}} \in \mathbb{R}^{N \times d_{\mathrm{subj}}}$ and object-specific feature map $\mathbf{Z}_{\mathrm{obj}} \in \mathbb{R}^{N \times d_{\mathrm{obj}}}$ are generated from $\mathbf{F} \in \mathbb{R}^{N \times d}$ as follows:

$$\mathbf{Z}_{\mathrm{subj}} = \mathbf{F}\mathbf{B}_{\mathrm{subj}} \quad \text{and} \quad \mathbf{Z}_{\mathrm{obj}} = \mathbf{F}\mathbf{B}_{\mathrm{obj}}. \tag{6}$$

### 2.4.1 Biometric Decoding

The biometric neural decoding is driven by a supervised multi-subject classification task. We apply a Global Average Pooling (GAP) operator, $\mathcal{G}_{\mathrm{subj}} : \mathbb{R}^{N \times d_{\mathrm{subj}}} \rightarrow \mathbb{R}^{d_{\mathrm{subj}}}$, to the subject-specific neural feature map $\mathbf{Z}_{\mathrm{subj}}$ to build a subject class token:

$$\mathbf{z}_{\mathrm{subj}}^{\mathrm{cls}} = \mathcal{G}_{\mathrm{subj}}(\mathbf{Z}_{\mathrm{subj}}). \tag{7}$$

Finally, a linear multi-subject classifier $\mathcal{C}_{\mathrm{subj}} : \mathbb{R}^{d_{\mathrm{subj}}} \rightarrow \mathbb{R}^{S}$ is applied for the final biometric prediction:

$$\hat{\mathbf{y}}_{\mathrm{subj}} = \mathcal{C}_{\mathrm{subj}}(\mathbf{z}_{\mathrm{subj}}^{\mathrm{cls}}). \tag{8}$$

### 2.4.2 Semantic Decoding

To establish a feature-level connection between the fMRI voxel signals $\mathbf{V}$ and the semantic content of visual stimulus $\mathbf{X}$, we utilize the vision feature map $\mathbf{F_x} \in \mathbb{R}^{N_x \times d_x}$ extracted from the last layer of a frozen CLIP visual encoder [33]. In contrast to most neural vision approaches that project fMRI features into the CLIP visual space [5, 38], our method takes the opposite strategy by treating the CLIP vision features as queries to extract corresponding neural object features from $\mathbf{Z}_{\mathrm{obj}}$.

This design is motivated by the complementary roles of CLIP and fMRI signals. CLIP features encode **subject-invariant** and **stimulus-driven** semantics with rich spatial and conceptual structure, effectively serving as a pseudo-ground-truth reference for object existence. In contrast, fMRI captures **subject-specific** neural responses that reveal how different individuals attend to these semantic components. To fuse these two modalities, we applied a multi-head cross-attention extractor $\mathcal{A} : \mathbb{R}^{N_x \times d_x} \times \mathbb{R}^{N \times d_{\mathrm{obj}}} \times \mathbb{R}^{N \times d_{\mathrm{obj}}} \rightarrow \mathbb{R}^{N_x \times d_{\mathrm{obj}}}$, defined as follows:

$$\mathbf{Z}_{\mathrm{obj}}^{\mathbf{F_x}} = \mathcal{A}\big(\mathrm{Query} = \mathbf{F_x}, \mathrm{Key} = \mathbf{Z}_{\mathrm{obj}}, \mathrm{Value} = \mathbf{Z}_{\mathrm{obj}}\big). \tag{9}$$

Here, the CLIP embeddings act as **queries**, while the fMRI-derived $\mathbf{Z}_{\mathrm{obj}}$ serves as **keys** and **values**. This configuration ensures that the fused representation remains fundamentally neural in nature: fMRI signals determine what semantic components are prioritized, while CLIP provides the structured semantic reference frame. The resulting cross-attention has two key effects:

- **Semantic Prioritization** - CLIP embeddings query the fMRI features, and attention weights highlight which parts of the CLIP semantic space align with neural activations. This allows the model to anchor predictions in semantically grounded content;

- **Subject-Specific Modulation** - The fMRI responses modulate CLIP-driven semantics in a subject-dependent manner, enabling the model to capture how different individuals selectively emphasize different attributes of visual stimuli that share similar semantic contents.

In this way, CLIP will not dominate or overwrite the neural signal, but rather provides a semantically structured scaffold. The fMRI features dynamically shape which aspects of that structure are emphasized, yielding a bi-directional synergy. Our semantic decoding thus remains primarily rooted in the fMRI modality, with CLIP assisting in refining object-specific neural features for final multi-label object classification. Similar to what we have done in the biometric decoding in the previous section, a global feature operator $\mathcal{G}_{\mathrm{obj}} : \mathbb{R}^{N_x \times d_{\mathrm{obj}}} \rightarrow \mathbb{R}^{d_{\mathrm{obj}}}$ transforms $\mathbf{Z}_{\mathrm{obj}}^{\mathbf{F_x}}$ into an object class token:

$$\mathbf{z}_{\mathrm{obj}}^{\mathrm{cls}} = \mathcal{G}_{\mathrm{obj}}(\mathbf{Z}_{\mathrm{obj}}^{\mathbf{F_x}}). \tag{10}$$

Following that a multi-label object classifier $\mathcal{C}_{\mathrm{obj}} : \mathbb{R}^{d_{\mathrm{obj}}} \rightarrow \mathbb{R}^{C}$ is applied for final semantic prediction:

$$\hat{\mathbf{y}}_{\mathrm{obj}} = \mathcal{C}_{\mathrm{obj}}(\mathbf{z}_{\mathrm{obj}}^{\mathrm{cls}}). \tag{11}$$

## 2.5 Model Training

The training process of our model consists of two stages. In the first stage, we follow the approach of SC-MBM [5] to pre-train a ViT-based masked autoencoder for fMRI data, constructing a latent neural space via minimizing reconstruction error with a Mean-Square Error (MSE) loss.

In the second stage, we retain only the fMRI encoder $\mathcal{E}(\cdot)$ from the first stage and optimize it with all other parameters in the proposed architecture. Three loss functions guide this stage. First, for biometric decoding, we introduce a subject classification loss $\mathcal{L}_{\text{subj}}$, which computes the cross-entropy $\mathcal{H}_{\text{CE}}$ with softmax activation against the one-hot label $\mathbf{y}_{\text{subj}}$ from the ground-truth subject index $\mathbf{y}_{\text{subj}}$:

$$\mathcal{L}_{\text{subj}} := \mathcal{H}_{\text{CE}}\big(\text{softmax}\left(\hat{\mathbf{y}}_{\text{subj}}\right), \mathbf{y}_{\text{subj}}\big). \tag{12}$$

For multi-label semantic decoding, we employ an object loss $\mathcal{L}_{\text{obj}}$ which is a binary cross-entropy function $\mathcal{H}_{\text{BCE}}$ with sigmoid activation $\sigma(\cdot)$:

$$\mathcal{L}_{\text{obj}} := \mathcal{H}_{\text{BCE}}\big(\sigma\left(\hat{\mathbf{y}}_{\text{obj}}\right), \mathbf{y}_{\text{obj}}\big). \tag{13}$$

Finally, we impose an orthonormal constraint on the learnable basis concatenation $\mathbf{B} = (\mathbf{B}_{\text{subj}}, \mathbf{B}_{\text{obj}})$, as defined in Eq. (6). This orthonormal loss $\mathcal{L}_{\text{orth}}$ ensures the perfect separation of subject-specific and object-specific features by minimizing $\mathcal{L}_{\text{orth}} := \|\mathbf{B}\mathbf{B}^\top - \mathbf{I}_d\|_{\mathbf{F}}^2$, where $\|\cdot\|_{\mathbf{F}}$ indicates the Frobenius matrix norm, and $\mathbf{I}_d \in \mathbb{R}^{d \times d}$ is the identity matrix of rank $d$.

In summary, in the second training stage, the total objective $\mathcal{L}$ is formulated as:

$$\mathcal{L} = \mathcal{L}_{\text{subj}} + \mathcal{L}_{\text{obj}} + \lambda \mathcal{L}_{\text{orth}}, \tag{14}$$

where $\lambda$ serves as a trade-off hyperparameter to balance the orthonormal constraint against the subject and object classification losses, which are considered equally important and share the same scale.

## 3 Experiments

### 3.1 Experimental Setup

**Dataset**. We evaluate our $i$MIND framework using the Natural Scenes Dataset (NSD) [2], a comprehensive, publicly available fMRI dataset capturing brain responses from 8 human subjects viewing natural scenes from MS-COCO [23]. Each subject passively viewed a set of 10,000 images for 3s, each repeated three times; 1,000 of these images were shared across all subjects, while the remaining 9,000 were unique to each individual, with no overlap between subjects. Due to incomplete sessions and data availability restrictions, not all trials are accessible for every subject, resulting in a total of 213,000 trials across all participants before pre-processing. In line with previous NSD studies [17, 35, 36, 38], we used standardized train/test splits and averaged fMRI activations over repetitions for each image within each subject. This pre-processing yielded 69,566 training samples and 7,674 test samples, allowing us to train a single multi-subject model across all 8 subjects. Additional details on NSD data, fMRI pre-processing, and $i$MIND implementation can be found in Appendix A.3.3.

### 3.2 Neural Decoding Performance

**Semantic Decoding**. For the semantic decoding task, we evaluate and compare with other models using three standard metrics for multi-label classification: mean Average Precision (mAP), the area under the receiver operating characteristic curve (AUC), and Hamming distance (Hamming). Table 1 categorizes models based on their ability to process multi-subject fMRI signals simultaneously or on a per-subject basis, as well as the modalities used for object classification. While our $i$MIND model is designed to process both image and fMRI modalities, it can also be adapted as a single-modality model by simply removing the cross-attention mechanism in Eq. (9) and using $\mathbf{Z}_{\text{obj}}$ in Eq. (6) directly for semantic decoding. Experimental results indicate that $i$MIND achieves superior performance across all three metrics, establishing a new state-of-the-art for semantic decoding in both single-modality and multi-modality settings.

Table 1: Semantic decoding performance on the NSD dataset.

| Model Type | Methods | Modalities | mAP ↑ | AUC ↑ | Hamming ↓ |
|---|---|---|---|---|---|
| Single-subject | MLP [44]* | fMRI | .258 | .854 | .033 |
| | ViT [44]* | fMRI | .238 | .815 | .032 |
| Multi-subject | MLP [44]* | fMRI | .150 | .767 | .039 |
| | ViT [44]* | fMRI | .156 | .755 | .038 |
| | EMB [4] | fMRI | .220 | .825 | .035 |
| | CLIP-MUSED [44] | fMRI+Image+Text | .258 | .877 | .030 |
| | *i*MIND (Ours) | fMRI | **.309** | **.913** | **.027** |
| | | fMRI+Image | **.784** | **.984** | **.012** |

\* directly sourced from [44] as benchmarks due to the limited research on semantic neural decoding

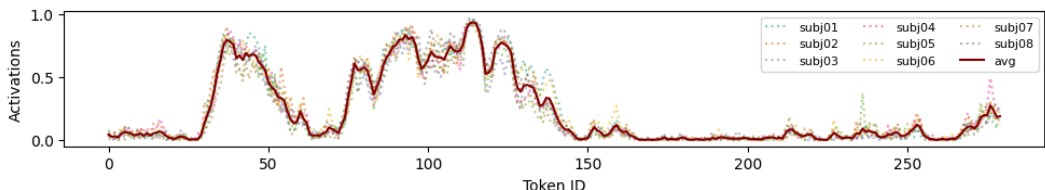

Figure 2: Average activations of tokens in $\mathbf{Z}_{\text{obj}}$ for the object *Chair*.

**Biometric Decoding**. To the best of our knowledge, no existing neural decoding models support subject classification using the fMRI modality. To provide a comprehensive and fair evaluation, we established baseline models following the exact preprocessing steps from our proposed method and conducted both supervised and unsupervised biometric decoding, as presented in Table 2. Top-1 accuracy (ACC) and Matthews Correlation Coefficient (MCC) [28] are used as metrics. The poor performance of naive subject classification methods underscores the com-

Table 2: Biometric decoding performance.

| Method | Setting | ACC | MCC |
|---|---|---|---|
| K-Means | Euclidean | .181 | .068 |
| | Cosine | .232 | .126 |
| MLP | Plain | .283 | .181 |
| | L2 norm | .377 | .290 |
| | L2 + ReLU | .573 | .526 |
| *i*MIND | – | **.999** | **.999** |

plexity of neural data across subjects, whereas the near-perfect classification achieved by *i*MIND highlights the effectiveness and necessity of our subject-object disentanglement approach. This demonstrates that within our framework, biometric fMRI features are highly discriminative across subjects and our linearity assumption is reasonable. By facilitating the extraction of task-relevant features, our disentanglement method further enhances downstream semantic and biometric decoding. Details on how we build biometric decoding baselines are provided in Appendix A.4.

### 3.3 Subject-Invariant Decoding

The primary motivation for the subject-object disentanglement design, introduced in Section 2.3, is to decompose the entangled neural feature $\mathbf{F}$ into subject-specific and object-oriented components $\mathbf{Z}_{\text{subj}}$ and $\mathbf{Z}_{\text{obj}}$ for a better biometric and semantic decoding. This approach expects object-wise token contributions in $\mathbf{Z}_{\text{obj}}$ to remain consistent across subjects. Using the object *chair* as an example, we visualize 280 tokens' activations averaged across all correct predictions by subject in Figure 2. It turns out that at the feature level, our method successfully achieves subject-invariant decoding, as token activations display high similarity with only negligible subject-level variations. This outcome demonstrates our model's effectiveness in extracting object-specific information from complex fMRI data, offering a robust framework for multi-subject fMRI decoding.

### 3.4 Visual-Neural Relationship

We empirically investigate the relationship between brain activities and semantic objects in visual stimuli, leveraging both GradCAM [37] and Attention Roll-out [1].

**Subject-wise 1D Activation Pattern**. Taking **subj01** as an example, we calculate voxel-wise activations within low-level visual regions of interest (ROIs) (V1-V4) and a wider high-level visual ROI in response to three objects: *person*, *horse*, and *chair*. This calculation takes the median activation across all true positive samples predicted by our model on the test set. As shown in Figure 3, the

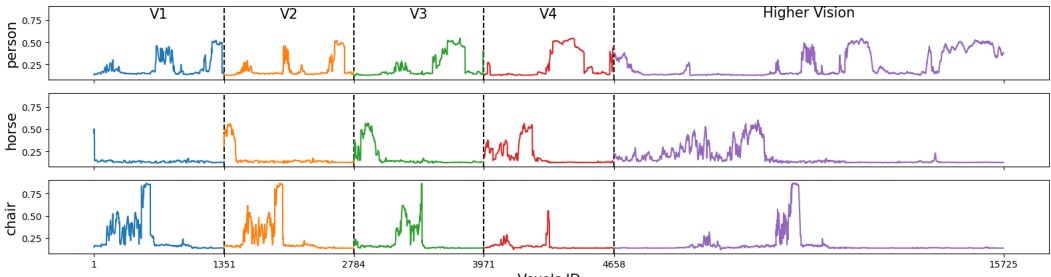

Figure 3: 1D Object-Voxel activations by brain vision ROIs for subj01.

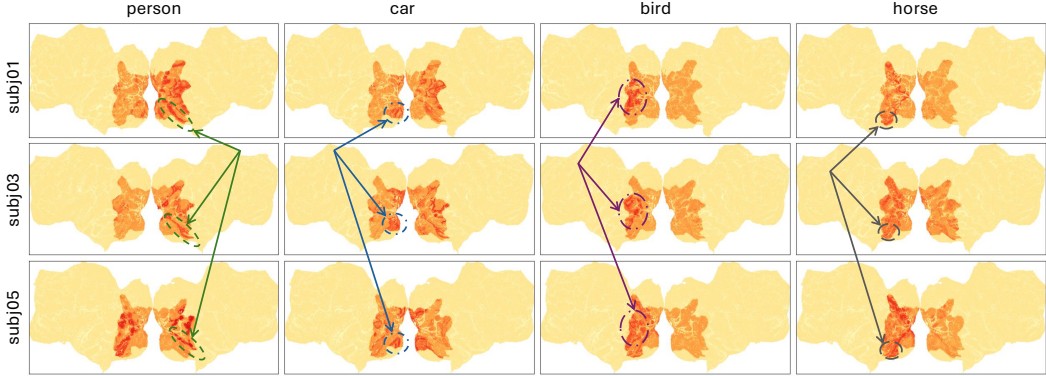

Figure 4: 3D Object-Voxel activations of *person*, *car*, *bird*, and *horse* for subj01, subj03, and subj05.

y-axis represents median activation, while the x-axis represents voxel IDs, providing a clear overview of the active voxels across both low-level and high-level visual ROIs when **subj01** recognizes each object. The unique activation patterns, with specific voxels responding to each object, indicate that brain voxels function differently from image pixels. In images, object locations are believed spatially random, so evaluating pixel-wise activation does not yield consistent spatial activation patterns. In contrast, fMRI voxels exhibit specialized roles in processing visual information, suggesting that brain voxels are organized by functional responsibility with a degree of spatial invariance–especially in high-level visual ROIs. This conclusion aligns with the existing studies from neuroscience [20, 21].

**Cross-subject 3D Activation Pattern**. We present 3D brain activation patterns in Figure 4 for four objects–*person*, *car*, *bird*, *horse*–visualized across three subjects: subj01, subj03, and subj05. Light yellow regions denote non-visual areas that were excluded from the dataset, resulting in uniformly absent signals in these regions. Based on the visualization, the following observations are made:

- **Consistency across subjects**: the objects *bird* and *horse* show a broad similarity across subjects, particularly in the region of the higher visual cortex and predominantly in the left hemisphere. This consistency suggests that certain high-level features associated with animals may be processed in similar ways across individuals, reflecting stable visual processing pathways in the brain.

- **Object sensitivity**: the activation intensity for object *person* appears stronger and more concentrated, indicating that the brain may allocate increased neural resources or "attention" to socially relevant stimuli (*people*), compared to less socially significant objects like *bird*. This result is supported by neuroscience research [15, 42].

- **Representational flexibility**: while general patterns are shared across subjects, the intensity and spatial distribution of activation vary slightly for certain objects, such as *car*. These variations may reflect individual differences in brain anatomy or prior experiences that influence object representation and visual information processing. This flexibility of the brain's adaptability to personal needs and experiences is known as neural plasticity [7, 11, 30].

### 3.5 Variations in Subject Attention

A key contribution of *i*MIND is its use of subject-invariant CLIP visual features to explore how different subjects focus on distinct objects when receiving the same stimulus. Figure 5 illustrates this attention variation: the first column displays the original visual stimulus with six ground-truth

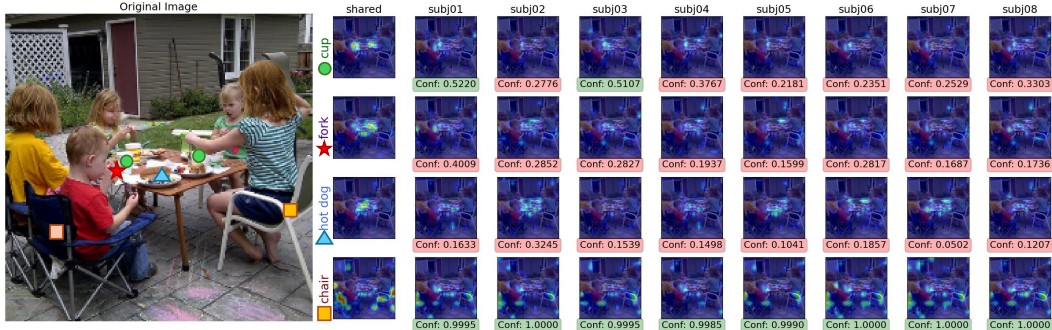

Figure 5: Variations in subjects' attention to different objects. The leftmost image shows the visual stimulus, labeled with six objects: *person*, *dining table*, *cup*, *fork*, *hot dog*, and *chair*. Four of them are selected for visualization. Plots in the second column represent the shared attention across all subjects, and the remaining eight columns show the residual, subject-specific attention alongside predicted probabilities to compare recognition confidence and priority.

annotations, the second column represents the shared attention map common across all subjects, and the remaining eight columns show the residual, subject-specific attention patterns. Four object-specific attention maps are visualized on rows with predicted logits to compare recognition confidence. Considering images are shown for only 3s [2], patterns of attention and object recognition offer even more intriguing insights into rapid, automatic processes of visual information and neural encoding.

**Temporal Constraints on Attention Allocation**: Within a brief 3-second viewing window, the brain must rapidly parse and prioritize elements of a complex scene. Notably, despite occupying only a modest portion of the image, the object *chair* consistently receives high attention across all subjects. This suggests that chairs are processed in an early, feed-forward manner–likely due to their high salience and distinctive visual features that enable rapid recognition under time constraints. To validate this, we analyzed prediction confidence for *chair* in the training set across subjects, grouped

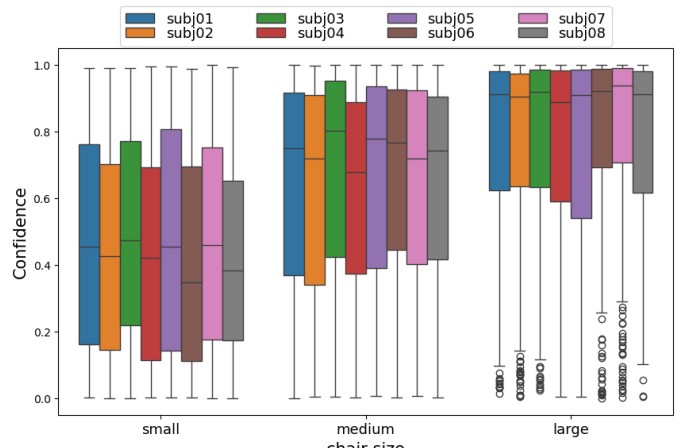

Figure 6: Recognition of *chair* by object size.

by object size (Figure 6). The results confirm that chairs of sufficient size are reliably identified by all participants. This immediate and confident response highlights the efficiency of the visual system in detecting familiar, contextually relevant objects with minimal cognitive effort.

**Subject-specific Focus Under Time Constraints**: Under these brief viewing conditions, subject-specific differences in attention to objects like *cup*, *fork*, and *hot dog* become especially revealing. Variation in attention to *cup*, particularly with subj01 and subj03 achieving recognition confidence (predicted probabilities > 0.5) by focusing more precisely on its location, suggests that these individuals may possess faster or more selective attentional strategies. Such patterns point to individual differences in visual processing speed, attentional control, or

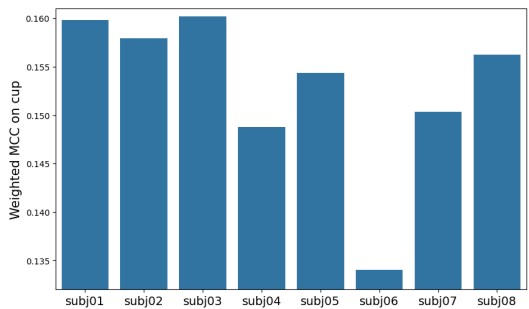

Figure 7: Sensitivity to *cup* across subjects.

perceptual expertise that influence object prioritization under time constraints. These findings are consistent with training results (Figure 7), where subj01 and subj03 show the highest sensitivity to cups, as measured by weighted MCC. Interestingly, although none of the subjects successfully recog-

nize *fork* or *hot dog* in Figure 5, all but subj02 allocate more attention to *fork* than *hot dog*, suggesting a subtle yet consistent attentional bias that reflects object familiarity or contextual relevance.

To the best of our knowledge, the proposed *i*MIND is the first model to capture subtle variations in how quickly and differently individuals allocate attention within a constrained time frame, demonstrating the model's robustness in simulating real-world neural processes. The model's ability to account for both shared and individual-specific attention patterns in response to brief stimulus exposure can inform the development of neural decoding approaches that better reflect human variability, especially in time-sensitive applications like real-time scene analysis or autonomous driving. Complete details for all visualized figures are provided in the Appendix.

Table 3: Ablation on loss functions.

| ID | $\mathcal{L}_{\text{subj}}$ | $\mathcal{L}_{\text{orth}}$ | $\lambda$ | mAP (%) |
|---|---|---|---|---|
| **Full** | ✓ | ✓ | **.1** | **78.36** |
| 1 | ✓ | | | -7.15 ($\downarrow$) |
| 2 | | ✓ | .1 | -11.17 ($\downarrow$) |
| 3 | | | | -11.07 ($\downarrow$) |
| 4 | ✓ | ✓ | .01 | -0.94 ($\downarrow$) |
| 5 | ✓ | ✓ | 1 | -2.95 ($\downarrow$) |
| 6 | ✓ | ✓ | 10 | -0.84 ($\downarrow$) |

In sum, the fact that subjects allocate attention differently within just a few seconds underscores the efficiency of neural mechanisms in prioritizing objects and the role of individual cognitive differences. This rapid, nuanced attention mapping highlights how our *i*MIND framework captures the interplay of shared and individual neural patterns, bridging cognitive neuroscience with computational modeling to decode visual attention in real-world scenarios.

### 3.6 Ablation Studies

**Loss Functions**. Our novel designs–subject-object disentanglement and the dual-decoding framework–are considered two key factors in achieving SOTA semantic decoding performance. To evaluate their effectiveness and necessity, we test combinations of the two loss functions, $\mathcal{L}_{\text{subj}}$ and $\mathcal{L}_{\text{orth}}$, along with a trade-off hyperparameter $\lambda$. Table 3 confirms that both $\mathcal{L}_{\text{orth}}$ for subject-object disentanglement and the dual-decoding design are crucial for achieving high semantic performance. Moreover, the trade-off parameter seems to have a minimal effect on the overall results. The optimal model utilizes all three loss functions with a trade-off parameter of $\lambda = 0.1$.

**Model Variants**. We investigate the impact of two key hyperparameters on the performance of semantic decoding: $d_{\text{obj}}$, the dimension of the neural object space for subject-object disentanglement in Section 2.3, and $h$, the number of heads in multi-head cross-attention module in Eq. (9). According to Table 4, increasing the number of heads does not necessarily lead to performance gain, as it may result in potential overfitting. In addition, we found that performance degradation remains minimal as long as there is sufficient feature space allocated for the neural object information. Ultimately, the optimal object classification performance, in terms of mAP, is achieved with $h = 4$ and $d_{\text{obj}} = 700$.

Table 4: Ablation on heads and $d_{\text{obj}}$.

| ID | Head(s) | $d_{\text{obj}}$ | mAP (%) |
|---|---|---|---|
| **Full** | **4** | **700** | **78.36** |
| 1 | **1** | 700 | -1.95 ($\downarrow$) |
| 2 | **2** | 700 | -1.12 ($\downarrow$) |
| 3 | **6** | 700 | -1.35 ($\downarrow$) |
| 4 | **8** | 700 | -9.18 ($\downarrow$) |
| 5 | 4 | **100** | -4.01 ($\downarrow$) |
| 6 | 4 | **200** | -2.11 ($\downarrow$) |
| 7 | 4 | **300** | -1.04 ($\downarrow$) |
| 8 | 4 | **400** | -1.38 ($\downarrow$) |
| 9 | 4 | **500** | -0.89 ($\downarrow$) |
| 10 | 4 | **600** | -1.14 ($\downarrow$) |

## 4 Conclusion

In this paper, we introduce an innovative multi-subject dual-decoding framework that decomposes latent fMRI representations into distinct subject-specific and object-specific components using a robust basis transformation. This approach enables precise biometric decoding through individualized neural features, while shared object-oriented features facilitate subject-invariant semantic decoding by querying with CLIP-derived visual representations. Our framework not only establishes a new benchmark for semantic decoding accuracy but also reveals variations in attentional focus across subjects when viewing identical visual stimuli. Additionally, we construct object-specific activation patterns at the voxel level, offering data-driven insights into the brain's visual processing mechanisms.

In future work, we aim to leverage large-scale fMRI datasets to develop more robust and informative pretrained models for extracting latent neural features. Additionally, we plan to collaborate with brain scientists to deepen our understanding of how specific voxel patterns in fMRI data relate to semantic object representations. This domain knowledge will help bridge the gap between visual features and neural signals, further enhancing the interpretability and accuracy of brain-based decoding models.

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

# A  Technical Appendices and Supplementary Material

## A.1  Related Work

### A.1.1  Single-Subject v.s. Multi-Subject Models

The application of deep learning to neural data has initially centered on single-subject models tailored to individual participants. BrainDiVE [24] adopts a generative approach, synthesizing images predicted to activate specific regions of the human visual cortex. Moving beyond visual modalities, Mind Reader [22] incorporates textual information to reconstruct complex images containing multiple objects from brain activities. Extending this multimodal approach, BrainSCUBA [25] takes advantage of contrastive vision-language models and large-language models to generate voxel-wise captions, eliminating the need for human-annotated voxel-caption data. While single-subject models have achieved notable success, they face inherent limitations. These models require large amounts of subject-specific data to train robust models, which is challenging given the high costs and effort involved in collecting fMRI data. Furthermore, they are prone to overfitting, exhibit poor generalizability across individuals, and struggle with scalability when applied to larger datasets or diverse populations.

To overcome these challenges, multi-subject models aim to unify data across participants, enabling shared representation learning. However, this approach introduces significant complexity due to inter-subject variability, which arises from static anatomical differences and dynamic functional responses. Various methods have been proposed to overcome these obstacles. [4] employs subject embeddings and recurrent architectures to account for inter-trial and inter-subject variability, outperforming many single-subject models in predicting MEG time series. MindBridge [41] introduces a biologically inspired aggregation function and a cyclic fMRI reconstruction mechanism to achieve subject-invariant representation learning. MindEye2 [36] aligns spatial patterns of fMRI activity to a shared latent space using subject-specific ridge regression, improving out-of-subject generalization with limited training data and achieving state-of-the-art results in image retrieval and reconstruction. More recently, CLIP-MUSED [44] introduced learnable subject-specific tokens to facilitate the aggregation of multi-subject data without a linear increase in model parameters. This approach integrates representational similarity analysis (RSA) to guide token representation learning based on the topological relationships of visual stimuli in the latent visual space.

These advancements demonstrate the potential of multi-subject models to surpass the limitations of single-subject approaches, providing more generalizable and scalable solutions for neural decoding tasks. However, to the best of our knowledge, existing multi-subject neural decoding models predominantly adopt what we term a **suppressive** strategy for handling inter-subject variability. This approach aims to minimize subject-specific differences during learning, progressively refining features to become more task-relevant as the model deepens. In these frameworks, subject-specific information is often treated as noise or an obstacle to effective decoding. In contrast, our $i$MIND framework proposes an **instructive** strategy. Rather than suppressing subject-specific differences, our model embraces this variability by explicitly disentangling subject-specific features from task-relevant ones. By doing so, $i$MIND not only preserves individual-specific neural representations but also leverages them positively to enhance both subject-specific and shared task-related decoding. This dual-decoding approach enables $i$MIND to achieve superior performance while offering insights into both individual neural patterns and shared semantic representations.

### A.1.2  Vision-Neural Interactions

Decoding visual information from neural signals is an inherently multi-modal task, involving the alignment and interaction of at least two modalities: images and neural signals (eg., fMRI). Broadly speaking, approaches for vision-neural modality alignment can be categorized into two branches based on the direction of projection between visual and neural spaces.

The first branch projects neural signals into a pre-trained latent visual space. This approach is exemplified by works such as [35], which maps flattened spatial patterns of fMRI activity across 3D cortical tissue cubes into the image embedding space of a pre-trained CLIP model. Similarly, [38] predicts latent representations of presented images from fMRI signals within the early visual cortex. Other notable works in this branch include [5, 22, 36, 41], which leverage pre-trained visual generative models, such as GANs [34] and diffusion models, for reconstruction tasks. These models capitalize

on large-scale visual datasets and avoid re-training resource-intensive generative architectures, which would otherwise be infeasible given the scarcity of paired neural-visual data. The second branch adopts the opposite approach by projecting latent visual image features into the neural space. This method is particularly useful for generating synthetic stimuli that activate specific brain regions, enabling the study of feature preferences in different areas of the brain. Classic examples include [18, 24, 25], which investigate neural activation patterns in response to synthetic stimuli derived from visual features.

Our $i$MIND model takes a fundamentally different approach to vision-neural modality interaction. Rather than projecting between modalities, we use CLIP-derived vision features as queries to extract corresponding neural object features directly from neural representations. This design choice is motivated by several factors. First, as a semantic neural decoding framework, $i$MIND does not rely on resource-intensive generative models. Second, direct projections between modalities often result in significant information loss and modality gaps that require careful handling. Most importantly, our approach is expected to enable the investigation of subject-specific attention variations when viewing the same visual stimuli. By treating the CLIP vision features as pseudo-ground-truths for object presence, we leverage their subject-invariant properties as an anchor to explore how neural responses to specific objects differ across subjects. This design uniquely aligns with the goals of understanding inter-subject variability in neural decoding.

## A.2 Theoretic Validation for Basis Transformation

In this section, we present the basis transformation in linear algebra and establish the relationship between coordinates in different bases. The derivation ensures clarity in transitioning from one basis to another, essential for interpreting subject-specific neural space $\mathcal{F}_{\text{subj}}$ and object-specific neural space $\mathcal{F}_{\text{obj}}$ mentioned at the end of Section 2.3 in the main paper. We begin with the following formal claim:

**Claim** Let $\mathcal{V}$ be a vector space of cardinality $d$ over $\mathbb{R}$, with a standard basis $\mathbf{E} = \{\mathbf{e}_1, \mathbf{e}_2, \ldots, \mathbf{e}_d\}$ of $\mathbb{R}^d$ and an arbitrary basis $\mathbf{B} = \{\mathbf{b}_1, \mathbf{b}_2, \ldots, \mathbf{b}_d\}$ of $\mathbb{R}^d$. For any vector $\mathbf{v} \in \mathcal{V}$, if its coordinate with respect to the standard basis $\mathbf{E}$ is given by $[\mathbf{v}]_{\mathbf{E}} \in \mathbb{R}^d$, then its coordinate with respect to the basis $\mathbf{B}$ can be derived as:

$$[\mathbf{v}]_{\mathbf{B}} = \mathbf{P}_{\mathbf{E} \to \mathbf{B}} \cdot [\mathbf{v}]_{\mathbf{E}}, \tag{15}$$

where $\mathbf{P}_{\mathbf{E} \to \mathbf{B}} \in \mathbb{R}^{d \times d}$ is the *change-of-basis* matrix from $\mathbf{E}$ to $\mathbf{B}$, defined as:

$$\mathbf{P}_{\mathbf{E} \to \mathbf{B}} := \begin{bmatrix} | & | & \cdots & | \\ \mathbf{b}_1 & \mathbf{b}_2 & \cdots & \mathbf{b}_d \\ | & | & \cdots & | \end{bmatrix}^{-1}. \tag{16}$$

**Proof** Since $\mathbf{v} \in \mathcal{V}$ and $\mathbf{E}$ forms a basis for the vector space $\mathcal{V}$, $\mathbf{v}$ can be written as a linear combination of all basis vectors from $\mathbf{E}$:

$$\mathbf{v} = \sum_{i=1}^{d} a_i \mathbf{e}_i, \tag{17}$$

where $a_i \in \mathbb{R}$ are scalars. In this case, the coordinate of $\mathbf{v}$ with respect to $\mathbf{E}$ is:

$$[\mathbf{v}]_{\mathbf{E}} = (a_1, a_2, \ldots, a_d)^\top \in \mathbb{R}^d. \tag{18}$$

Similarly, because $\mathbf{B}$ also forms a basis for $\mathcal{V}$, we can express $\mathbf{v}$ as:

$$\mathbf{v} = \sum_{i=1}^{d} w_i \mathbf{b}_i, \tag{19}$$

where $w_i \in \mathbb{R}$ are scalars. The coordinate of $\mathbf{v}$ with respect to $\mathbf{B}$ is:

$$[\mathbf{v}]_{\mathbf{B}} = (w_1, w_2, \ldots, w_d)^\top \in \mathbb{R}^d. \tag{20}$$

Since each basis vector $\mathbf{e}_i$ within $\mathbf{E}$ is also an element of the vector space $\mathcal{V}$, it can also be written as a linear combination of all basis vectors from $\mathbf{B}$:

$$\mathbf{e}_i = \sum_{j=1}^{d} p_{ji}\mathbf{b}_j. \tag{21}$$

Writing the equation above in matrix form, we obtain:

$$\begin{bmatrix} | & | & \cdots & | \\ \mathbf{e}_1 & \mathbf{e}_2 & \cdots & \mathbf{e}_d \\ | & | & \cdots & | \end{bmatrix} = \begin{bmatrix} p_{11} & p_{12} & \cdots & p_{1d} \\ p_{21} & \ddots & & p_{2d} \\ \vdots & & \ddots & \vdots \\ p_{d1} & p_{d2} & \cdots & p_{dd} \end{bmatrix} \begin{bmatrix} | & | & \cdots & | \\ \mathbf{b}_1 & \mathbf{b}_2 & \cdots & \mathbf{b}_d \\ | & | & \cdots & | \end{bmatrix}. \tag{22}$$

Denoting the middle matrix as $\mathbf{P}$ and solving it, we get:

$$\mathbf{P} := \begin{bmatrix} p_{11} & p_{12} & \cdots & p_{1d} \\ p_{21} & \ddots & & p_{2d} \\ \vdots & & \ddots & \vdots \\ p_{d1} & p_{d2} & \cdots & p_{dd} \end{bmatrix} = \begin{bmatrix} | & | & \cdots & | \\ \mathbf{b}_1 & \mathbf{b}_2 & \cdots & \mathbf{b}_d \\ | & | & \cdots & | \end{bmatrix}^{-1}. \tag{23}$$

Next, let's plug each $\mathbf{e}_i$ into Eq. (17) using the formulation of Eq. (21):

$$\mathbf{v} = \sum_{i=1}^{d} a_i \mathbf{e}_i = \sum_{i=1}^{d} (a_i \sum_{j=1}^{d} p_{ji}\mathbf{b}_j) = \sum_{j=1}^{d}(\sum_{i=1}^{d} a_i p_{ji})\mathbf{b}_j. \tag{24}$$

Combining with Eq. (19), we have:

$$\sum_{j=1}^{d} w_j \mathbf{b}_j = \sum_{j=1}^{d}(\sum_{i=1}^{d} a_i p_{ji})\mathbf{b}_j. \tag{25}$$

Move everything to the left hand:

$$\sum_{j=1}^{d}[w_j - (\sum_{i=1}^{d} a_i p_{ji})]\mathbf{b}_j = 0. \tag{26}$$

According to the claim, $\mathbf{B} = \{\mathbf{b}_1, \mathbf{b}_2, \ldots, \mathbf{b}_d\}$ is a basis of $\mathbf{R}^d$. Therefore, Eq. (26) holds if and only if:

$$w_j - \sum_{i=1}^{d} a_i p_{ji} = 0 \quad \text{for } j = 1, 2 \ldots, d. \tag{27}$$

Equivalently, we have:

$$\begin{bmatrix} w_1 \\ w_2 \\ \vdots \\ w_d \end{bmatrix} = \sum_{i=1}^{d} a_i \begin{bmatrix} p_{1i} \\ p_{2i} \\ \vdots \\ p_{di} \end{bmatrix} = \begin{bmatrix} p_{11} & p_{12} & \cdots & p_{1d} \\ p_{21} & \ddots & & p_{2d} \\ \vdots & & \ddots & \vdots \\ p_{d1} & p_{d2} & \cdots & p_{dd} \end{bmatrix} \begin{bmatrix} a_1 \\ a_2 \\ \vdots \\ a_d \end{bmatrix}. \tag{28}$$

Using the coordinates expression defined in Eq. (18) and Eq. (20) along with Eq. (23), we obtain the following equation:

$$[\mathbf{v}]_{\mathbf{B}} = \mathbf{P} \cdot [\mathbf{v}]_{\mathbf{E}} \quad \text{where } \mathbf{P} = \begin{bmatrix} | & | & \cdots & | \\ \mathbf{b}_1 & \mathbf{b}_2 & \cdots & \mathbf{b}_d \\ | & | & \cdots & | \end{bmatrix}^{-1}. \tag{29}$$

Finally, we complete the proof of the claim.

In our $i$MIND model, subject-object disentanglement is achieved through a basis transformation described above, where the new basis $\mathbf{B}$ is treated as learnable parameters optimized by loss propagation. We further enforce the orthonormality constraint on $\mathbf{B}$, as this ensures that any subspace spanned by a subset of $\mathbf{B}$ is orthogonal (complementary) to its counterparts in the original feature space. Specifically, in our model, this constraint guarantees that the subject-specific neural space $\mathcal{F}_{\text{subj}}$, spanned by $\mathbf{B}_{\text{subj}}$, and the object-specific neural space $\mathcal{F}_{\text{obj}}$, spanned by $\mathbf{B}_{\text{obj}}$, are complementary and non-overlapping. Consequently, this orthogonal decomposition yields a perfect separation of subject-specific and object-specific neural features within the latent neural representations.

### A.3 NSD dataset, Pre-processing, and Implementation

#### A.3.1 NSD Dataset

The Natural Scenes Dataset (NSD) [2] is a groundbreaking resource in cognitive neuroscience and artificial intelligence, designed to capture extensive, high-resolution fMRI data during natural scene perception. It includes whole-brain fMRI measurements of eight human participants at 7T field strength, with a spatial resolution of 1.8 mm. Participants viewed a total of $70,566$ natural scene images, with $10,000$ unique images per subject (9,000 unique to each participant and $1,000$ shared across all participants). Images were sourced from the richly annotated Microsoft COCO dataset [23], ensuring ecological relevance and diversity. The experiment was conducted over 30–40 sessions per participant, taking a rapid event-related design with continuous recognition tasks to guarantee engagement and probe both short- and long-term memory processes. During neural recording, participants were tasked with identifying objects in the image, with each visual stimulus presented for only three seconds per trial. This design makes the NSD particularly well-suited for investigating the mechanisms of rapid attention and visual recognition in human vision. Advanced pre-processing completed by the authors, including denoising and voxel-specific hemodynamic response modeling, yielded high-quality single-trial beta estimates with exceptional signal-to-noise ratios. Complementing the functional data, NSD includes extensive anatomical scans, resting-state data, and behavioral performance measures, enabling multi-faceted investigations of vision and memory. This dataset, with its unparalleled scale and quality, serves as a valuable benchmark for developing and testing machine learning models that aim to decode brain activity and simulate neural representations of natural scenes. In addition, the NSD dataset supports multiple widely used neuroimaging atlases to facilitate data analysis and integration with existing frameworks. Functional data are provided in both native cortical surface space and standard volumetric spaces, including fsaverage [13] and MNI152 [31], enabling compatibility with tools like FreeSurfer [12] and FMRIB Software Library. Additionally, the dataset includes manually defined regions of interest (ROIs) for retinotopic mapping and category-selective areas, such as the early visual cortex and higher-order regions in the ventral visual stream, which is the atlas that we used in our *i*MIND model. These comprehensive atlases allow researchers to seamlessly apply NSD data to diverse analytic pipelines and cross-study comparisons.

#### A.3.2 Preprocessing

Our preprocessing pipeline begins with splitting the dataset into training and testing sets. Due to incomplete sessions and data availability constraints, not all trials are accessible for every subject, resulting in a total of $213,000$ trials across all participants. Among these, neural recordings corresponding to $1,000$ images viewed by all subjects are allocated to the testing set, comprising a total of $21,118$ test trials. The remaining neural recordings, corresponding to images viewed exclusively by individual subjects, are included in the training set, resulting in $191,882$ training trials. Both training and testing trials are standardized voxel-wise using the mean and standard deviation calculated from the training set. Since each image is presented to a subject three times, we average the fMRI responses across repetitions for each image within each subject. This results in $69,566$ training samples and $7,674$ testing samples, allowing us to train and evaluate a single multi-subject model across all eight subjects. For each sample, we use the *nsdgeneral* atlas provided by the NSD dataset to extract visual voxel signals as a 1D vector. However, the number of visual voxels varies between subjects due to anatomical differences, with the voxel length $L_s$ ranging from $12,682$ to $17,907$ across subjects. To unify the input length, we apply a padding strategy inspired by Mind-Vis [5], which conducts wrap-around padding. This approach avoids issues arising from truncation or constant padding to the maximum voxel length. Additionally, since our fMRI encoder is based on a Vision Transformer (ViT), which requires input lengths divisible by the user-defined patch size (64 in our model), we adjust the uniform voxel length $L$ accordingly. The final voxel length across subjects is set to $L = 17,920$, ensuring compatibility with the model while maintaining consistency across participants.

#### A.3.3 Implementation Details

As described in Section 2 of the main paper, our proposed architecture is trained in two stages. The first stage involves pre-training a ViT-based masked autoencoder, similar to SC-MBM [5], using a self-supervised fMRI reconstruction task. In this stage, we choose a patch size of 64 voxels with a masking ratio of 0.75. The encoder has a hidden dimension of 768 and consists of 12 layers

Table 5: Generalizability to unseen subjects

| ID | Trained on | Tested on | mAP (%) |
|---|---|---|---|
| M7 | subj01–07 | subj01–07 | .7904 |
|  | subj01–07 | subj08 | .7842 |
| M8 (**Full**) | subj01–08 | subj01–07 | .7842 |
|  | subj01–08 | subj08 | .7909 |

of 6-head self-attention, while the decoder has a hidden dimension of 512 and 8 layers of 8-head self-attention. In the second stage, we discard the decoder and inherit only the encoder from the first stage, which outputs pre-trained feature $\mathbf{F} \in \mathbb{R}^{N \times d}$ with $N = 280$ and $d = 768$. We set the object neural space dimension $d_{\text{obj}} = 700$ and choose a 4-head cross-attention module for fMRI-vision feature interactions. The CLIP visual encoder we used is clip-vit-base-patch16 released by OpenAI, which remains frozen at all stages of the proposed framework. A trade-off parameter $\lambda$ of 0.1 is set by default to enforce the orthonormal constraint of the learnable basis $\mathbf{B}$. A detailed investigation is provided in Section 3.6. During this stage, all parameters are optimized end-to-end for subject and object classification tasks. For either stage, we train the model for 100 epochs, including 10 warm-up epochs. The learning rate is initialized at $7.5 \times 10^{-4}$ and terminated at zero adjusted dynamically by a cosine scheduler. The batch size is set to 200. Optimization is performed via the AdamW optimizer with a weight decay of 0.05. All experiments are conducted on two Nvidia RTX 6000 Ada GPUs, with the first stage taking approximately 1.5 hours and the second stage around 2 hours to complete.

### A.4 Subject Classification Baselines

To the best of our knowledge, no existing models support subject classification. To provide a comprehensive evaluation, we established baseline models using the exact fMRI preprocessing steps in Appendix A.3.2 and conducted both supervised and unsupervised biometric decoding. All methods are trained and tested on identical data splits and fMRI voxel sets as iMIND, ensuring a fair comparison on the same held-out unseen test set.

For supervised learning, we employ a single linear layer trained in two ways:

- Linear Regression: We minimize the mean squared error (MSE) between the input (padded fMRI voxel signals) and the target (one-hot subject IDs), using the ordinary least squares closed-form solution;

- Classification: We train an identical architecture with cross-entropy loss, treating subject identification as a standard classification task.

For unsupervised learning, we evaluate K-Means clustering with two distance metrics:

- Euclidean (L2) distance;

- Cosine similarity.

Since the number of subjects is known (8), we set the number of clusters to 8 as well. To measure performance, we compute accuracy and MCC by optimally aligning the learned clusters with ground-truth subject IDs.

### A.5 Subject Generalizability

For completeness, we conducted an additional experiment to assess subject generalizability within NSD as shown in Table 5. In our original setup (denoted as M8), iMIND was trained on data from all 8 subjects. For this experiment, we introduced a variant (M7), where iMIND was trained using only the first 7 subjects and tested on the held-out data of subj08. **M7/M8** achieved an overall mAP of **.7904/.7842** on the first 7 subjects and **.7842/.7909** on subj08. These results demonstrate that our proposed method exhibits a reasonable and strong generalizability in neural signal semantic decoding, particularly for unseen subjects.

### A.6 Limitations

While our method achieves state-of-the-art performance in both semantic and biometric decoding tasks, several limitations remain unresolved.

First, the current approach to neural feature extraction may not be optimal. Although functional, the pretrained neural reconstruction stage produces fMRI reconstructions–both numerically and visually–that underperform compared to the ground-truth voxel signals. This suggests that the masked autoencoder (MAE) backbone may not be the most effective architecture for this task, warranting further exploration.

Second, flattening voxel inputs discards crucial spatial relationships among neighboring voxels, despite neuroscientific evidence that proximal voxels exhibit functional coupling in visual processing. Future work could explore advanced architectures–such as 3D SwinTransformers, which are explicitly designed for volumetric fMRI data and have demonstrated efficacy in neurological disease diagnosis– to better preserve spatial hierarchies and improve feature learning.

Third, we acknowledge that the brain's functional dynamics are fundamentally non-linear and complex. Our assumption that subject-specific and object-specific components are linearly entangled at the latent feature level is a simplifying inductive bias introduced to enable interpretable, computationally tractable disentanglement. We agree it fully captures the richness of brain representations; rather, it serves as a first-order approximation that enables clear factorization of subject identity and semantic content from fMRI signals. If the linearity assumption does not fully hold, we expect the following potential implications:

- If subject-object interactions are fundamentally non-linear, the disentangled object representation $\mathbf{Z}_{obj}$ may still retain residual subject-specific information, potentially introducing subject bias in semantic decoding and diminishing our model's generalizability for unseen subjects.
- Conversely, enforcing strict linear disentanglement may suppress relevant non-linear object features in $\mathbf{Z}_{obj}$, potentially smoothing out sharp voxel-object modulations or degrading decoding performance for fine-grained categories.

Last, our analysis of visual mechanisms relies on post hoc interpretation methods (Grad-CAM and Rollout), which provide only approximate explanations of model behavior. A more principled approach would involve explainable-by-design architectures for fMRI feature extraction, which we leave for future work.

### A.7 Broader Impacts

Our work represents a pioneering step toward decoding the brain's visual processing mechanisms, with far-reaching implications for both neuroscience and artificial intelligence. By modeling how the brain transforms visual signals into neural activity and high-level semantics, we aim to uncover the fine-grained functional organization of visual regions–such as those specialized for distinguishing closely related objects (e.g., dogs vs. cats).

This understanding could enable breakthroughs in brain-computer interfaces (BCIs), where precise neural decoding could restore or augment vision for impaired individuals. Conversely, it also raises ethical considerations: the same principles could theoretically be used to manipulate neural signals, artificially inducing semantic perceptions (e.g., generating "fake" visual concepts in the brain). Such capabilities would necessitate rigorous ethical frameworks to prevent misuse while maximizing societal benefit.

Further, our computational approach bridges AI and neuroscience, offering interpretable models that could inspire more biologically plausible machine vision systems. By aligning artificial and biological vision, we may accelerate progress in both fields–from improving AI's robustness to advancing treatments for neurological disorders.

### A.8 Technical Details on Object-Voxel Visualization

In our experiments, we empirically investigate the relationship between brain activities and semantic objects in visual stimuli. In this section, we detail how voxel contributions to object recognition are

visualized in our framework. Given an input fMRI voxel signal $\mathbf{V} \in \mathbb{R}^L$, we obtain its object-specific neural feature map $\mathbf{Z}_{\text{obj}} \in \mathbb{R}^{N \times d_{\text{obj}}}$ before cross-attention module in our model. Using GradCAM [37], we are able to build an activation map $\mathbf{t} \in \mathbb{R}^N$, which quantifies the contribution of each neural token $\mathbf{z}_{\text{obj}} \in \mathbb{R}^{d_{\text{obj}}}$ in object-specific neural feature map $\mathbf{Z}_{\text{obj}} \in \mathbb{R}^{N \times d_{\text{obj}}}$ to the correct recognition of the object of interest.

Unlike CNN-based models, which can simply resize $t$ to match the size of the input $\mathbf{V}$ because of their spatial invariance nature (a one-to-one correspondence between input patches and latent feature vectors), our ViT-based neural encoder in $i$MIND model lacks such spatial invariance properties by default. For instance, the first neural token in $\mathbf{Z}_{\text{obj}}$ does not necessarily correspond to the first voxel patch of $\mathbf{V}$, making direct resizing infeasible.

To address this, we leverage Attention Roll-out [1] to approximate how information flows from the input voxels $\mathbf{V}$ to the neural tokens in feature map $\mathbf{Z}_{\text{obj}}$. Specifically, this information flow in our ViT-based encoder can be measured as follows:

$$\mathbf{A}^l = \mathbf{A}^{(l)} \mathbf{A}^{(l-1)} \cdots \mathbf{A}^{(2)} \mathbf{A}^{(1)}, \tag{30}$$

where $\mathbf{A}^{(k)} \in \mathbb{R}^{N \times N}$ represents the attention weights in the $k$-th self attention layer of the ViT encoder. Based on the mathematical property of the self-attention mechanism, the element $a_{ij}$ of the attention weights $\mathbf{A}$ at every transformer block defines how much attention flows from the token $j$ in the previous layer to the token $i$ in the next layer. Therefore, each element $a_{ij}^l$ within $\mathbf{A}^l$ defined in Eq. (30) quantifies the degree of information flow from the $j$-th voxel patch of input fMRI signal $\mathbf{V} \in \mathbb{R}^L$ to the $i$-th token in the feature map $\mathbf{Z}_{\text{obj}}$.

Next, we combine the GradCAM-based token contributions $\mathbf{t} \in \mathbb{R}^N$ with the cumulative information flow $\mathbf{A}^l$ to derive the voxel-level activation measurement $\mathbf{T}$:

$$\mathbf{T} = \mathbf{t} \cdot \mathbf{A}^l \in \mathbb{R}^N. \tag{31}$$

Here, $\mathbf{T}$ measures the contribution of each voxel path immediately after embedding the original 1D voxel signal $\mathbf{V}$ of length $L$ into $N$ patches. Since $\mathbf{T}$ is now positionally aligned with the input voxel patches, it can be safely upsampled from size $N$ to $L$ to obtain a voxel-level activation map for each fMRI sample. Unfortunately, this upsampled activation map is partially synthetic because wrap-around padding was applied during preprocessing to achieve a uniform, model-compatible voxel length $L$ across subjects. This padding introduces artificial "fake" voxels. The advantage of the wrap-around padding strategy is that it allows us to trace the origins of these fake voxels. To restore the true voxel activation map, we retain the activations of the real voxels, while for the fake voxels, we trace their origins and assign their values as the maximum of the original and artificial activations. This approach ensures that the restored activation map accurately represents the contributions of real voxels while mitigating the impact of synthetic padding.

Finally, this process is repeated across all samples containing the object of interest, enabling us to investigate voxel semantic selectivity, as illustrated in Figure 3 of the main paper. To achieve the 3D activation like Figure 4 in the main paper, we can just map each flattened voxel back to 3D brain space using the provided *nsdgeneral* atlas. This methodology allows us to map neural activations back to their voxel-level origins, providing insights into the relationship between neural representations and object recognition.

### A.9   Technical Details on Figure 6

We first computed the pixel occupation ratio for all training images containing the object *chair*. This ratio was derived by dividing the number of chair pixels (using MS-COCO annotations for masking) by the total image resolution. Since the raw pixel ratios exhibited a highly skewed distribution, we applied a log transformation to approximate a normal-like distribution as shown in Figure A.9.

Next, we calculated the mean $\mu$ and standard deviation $\sigma$ of the log-transformed ratios. To partition the chairs into size-based categories, we defined three intervals:

- Small chairs: $(-\infty, \mu - 0.5\sigma)$
- Medium chairs: $(\mu - 0.5\sigma, \mu + 0.5\sigma)$
- Large chairs: $(\mu + 0.5\sigma, 0)$

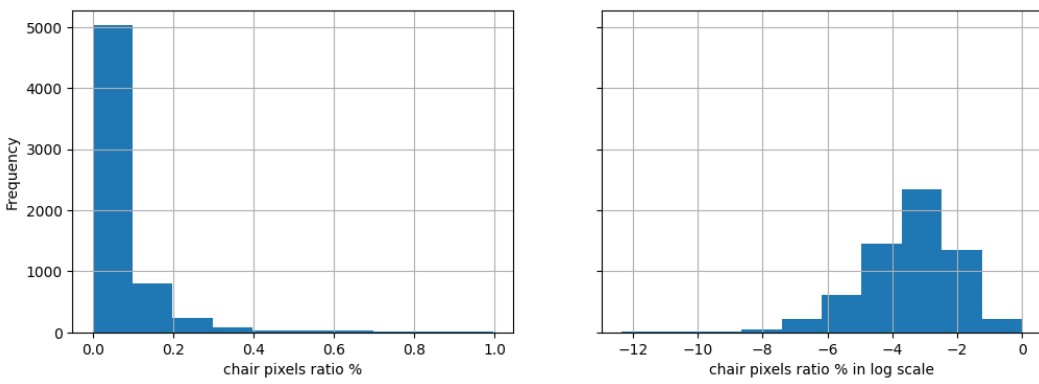

Figure 8: *Chair* pixel ratio distribution in training set

Finally, for each subject, we generated a boxplot of the predicted probabilities for the *chair* class, stratified by these size groups.

## A.10  Technical Details on Figure 7

To analyze differences in subjects' sensitivity to the object *cup*, we first computed the pixel occupation ratio for all training images containing *cup* and calculated the baseline MCC subject by subject as the pre-adjusted sensitivity measure. Since each subject viewed distinct images during training, we accounted for distribution shifts in both the size and frequency of cup appearances across subjects. To ensure a fair comparison, we adjusted the MCC by normalizing it with the subject-wise average pixel ratio. The resulting weighted MCC is visualized in Figure 7.

## A.11  More Analytical and Visual Results

### A.11.1  Voxel Sensitivity

To examine voxel sensitivities, we analyzed the mean (x-axis) and standard deviation (y-axis) of voxel activations across five brain ROIS for objects *bench* and *chair*, as presented in Figure A.11.1. The results indicate a quadratic relationship in voxel sensitivity across these regions, allowing us to classify voxels into three distinct groups based on their mean activation and variability (standard deviation). Each group reflects a unique role in semantic decoding within visual regions:

- Bystanders – This group, characterized by the lowest mean and standard deviation, consists of voxels that consistently contribute minimal information to the semantic decoding of visual stimuli. These voxels are either not responsible for distinguishing specific objects (*bench* and *chair*) or likely located in regions less involved in object discrimination, and instead providing generalized but stable responses across diverse stimuli.

- Discriminators – This higher mean and the highest standard deviation group includes voxels that show selective, highly variable responses, playing a key role in differentiating between features and supporting object-specific sensitivity. These voxels likely drive the flexibility needed for nuanced and accurate decoding of semantic information in visual stimuli.

- Supporters – The highest mean, low standard deviation voxels, characterized by strong, consistent activation, likely represent core object features and provide a stable foundation for robust and invariant decoding across subjects. They likely provide stable, foundational support for correctly classifying objects across different conditions.

These findings suggest that voxel sensitivity patterns vary across the visual hierarchy, with each group contributing distinct vision information-processing roles in object recognition in the brain.

### A.11.2  Single-subject 1D Activation Pattern

Similar to Figure 3 in the main paper, we provide more visualization results on 1D Object-Voxel activation below:

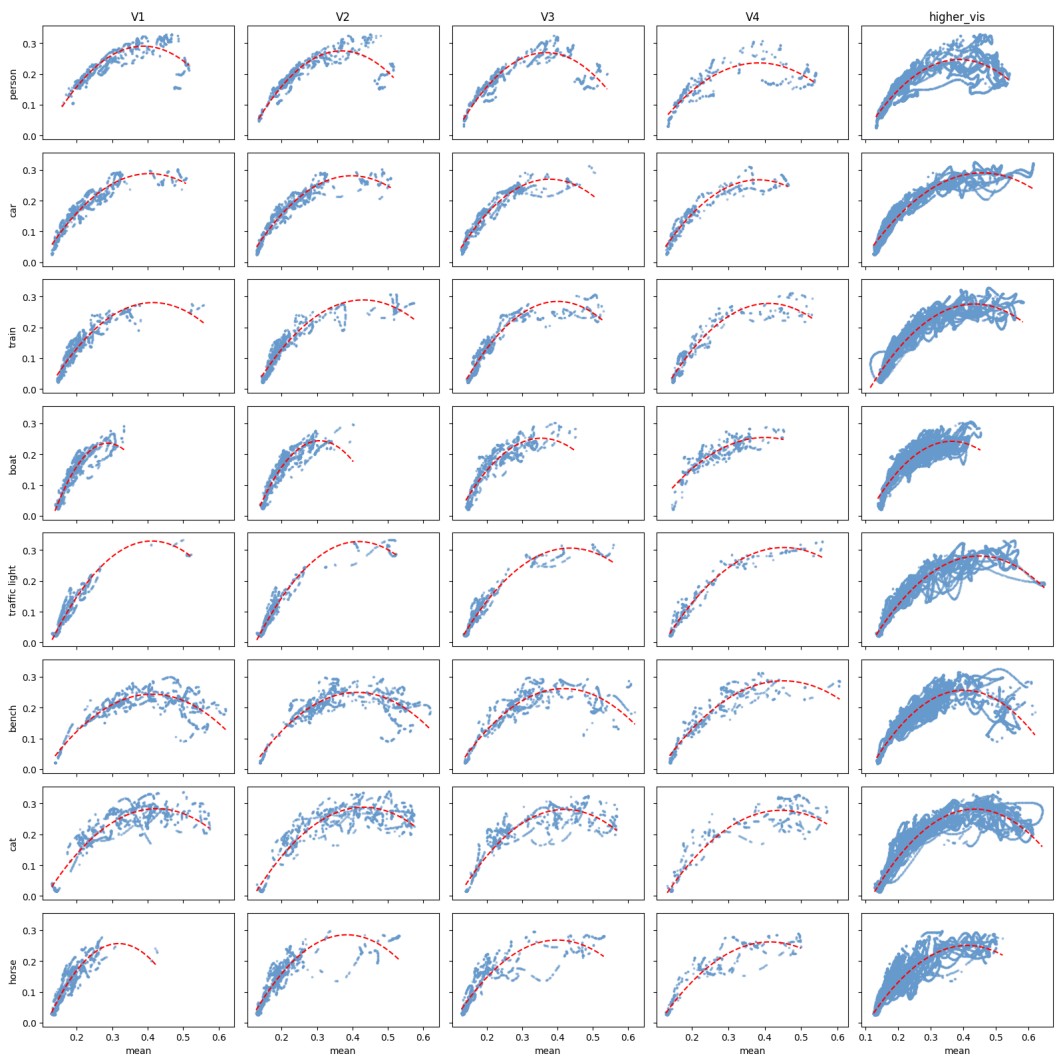

Figure 9: Object-Voxel activations (std v.s. mean) by vision ROIs for subj01

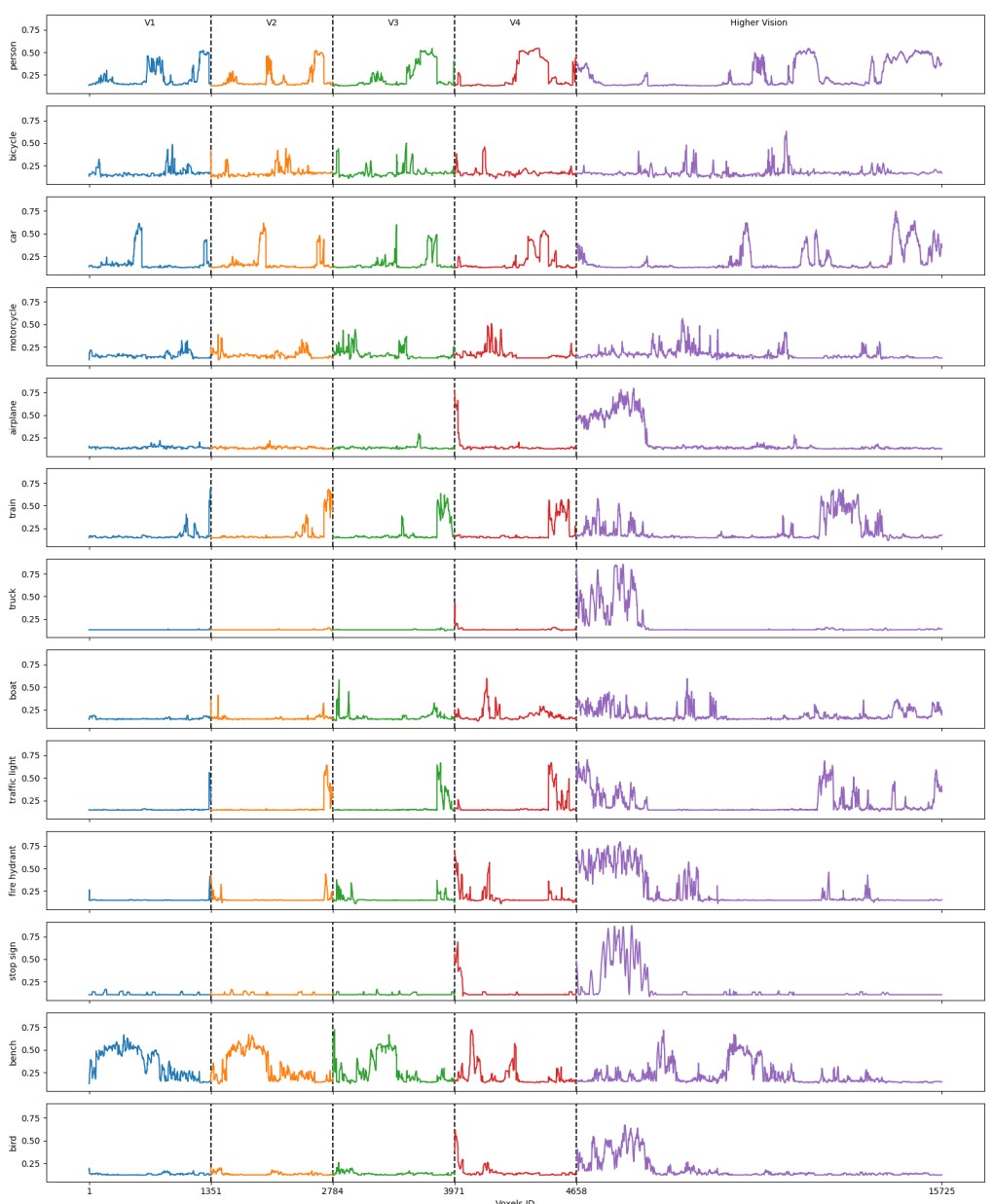

Figure 10: 1D Object-Voxel activations by brain vision ROIs for subj01.

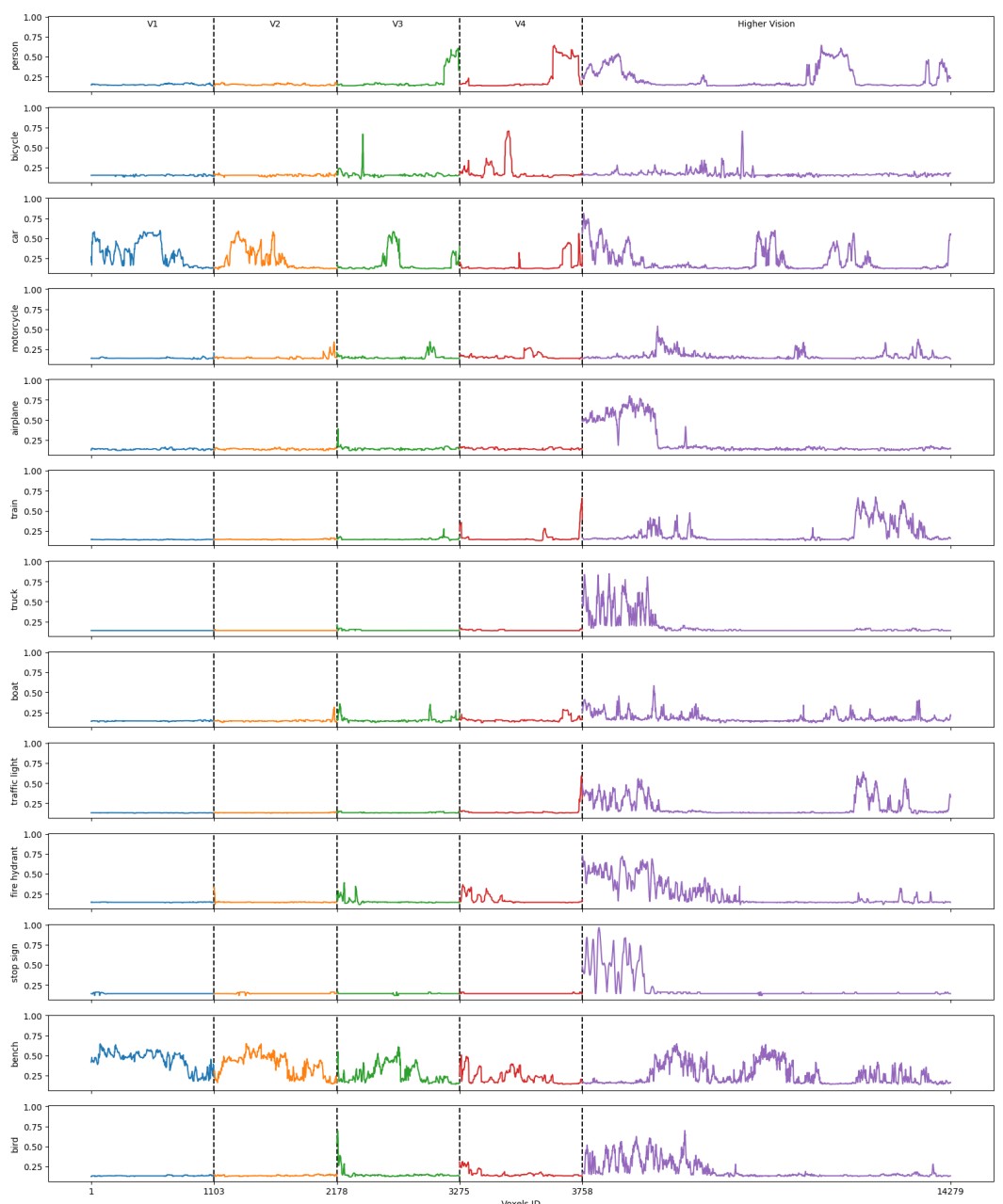

Figure 11: 1D Object-Voxel activations by brain vision ROIs for subj02.

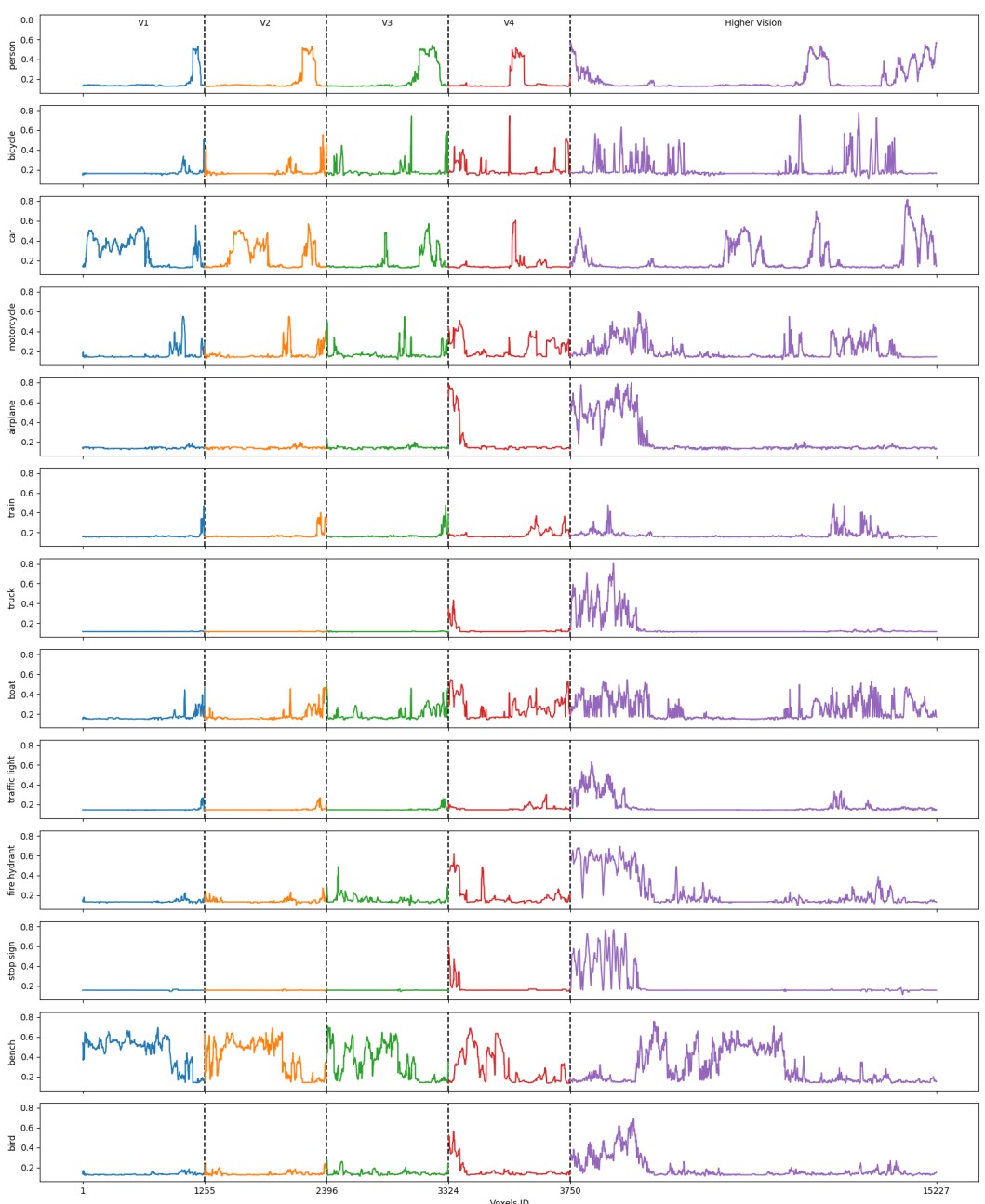

Figure 12: 1D Object-Voxel activations by brain vision ROIs for subj03.

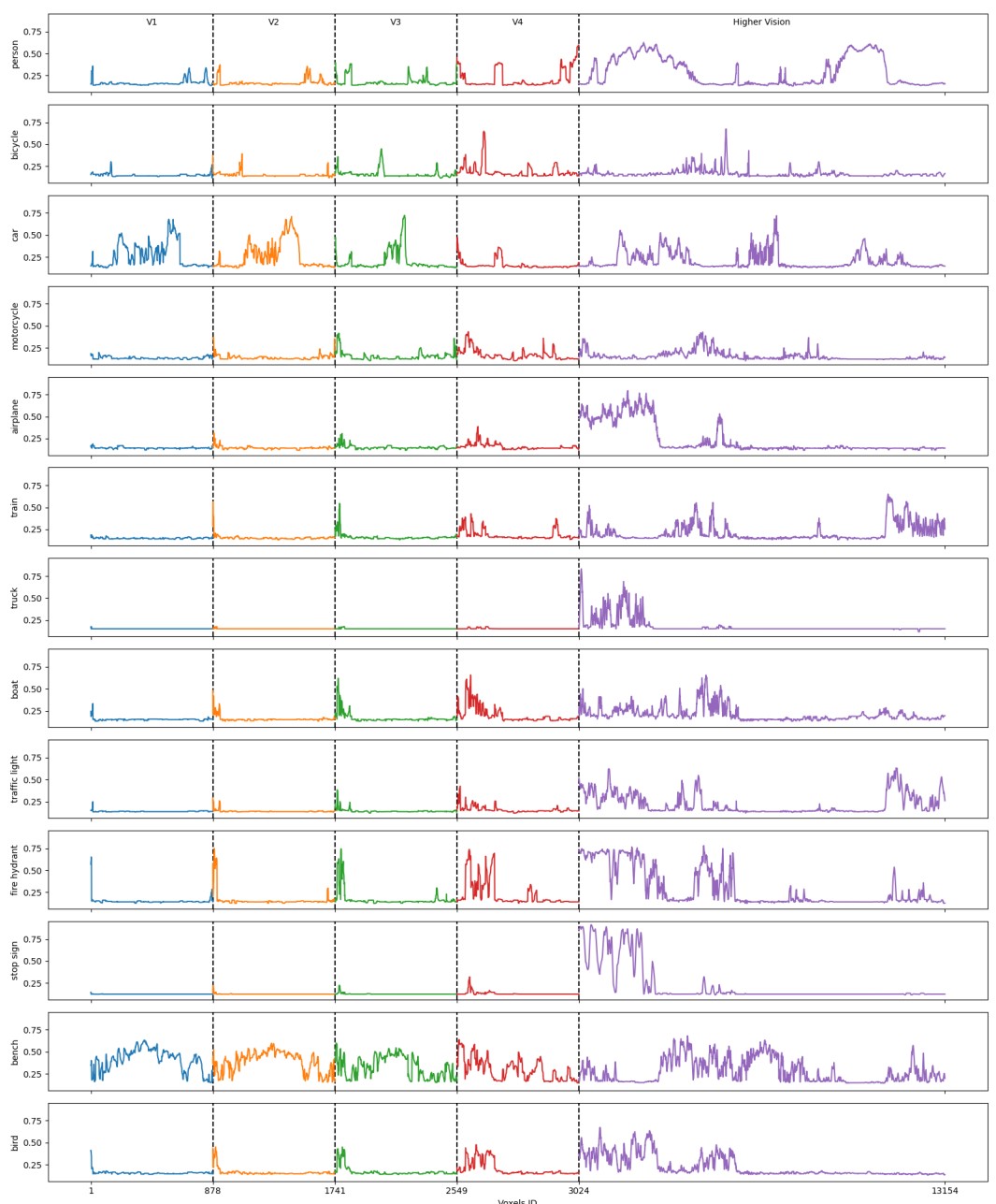

Figure 13: 1D Object-Voxel activations by brain vision ROIs for subj04.

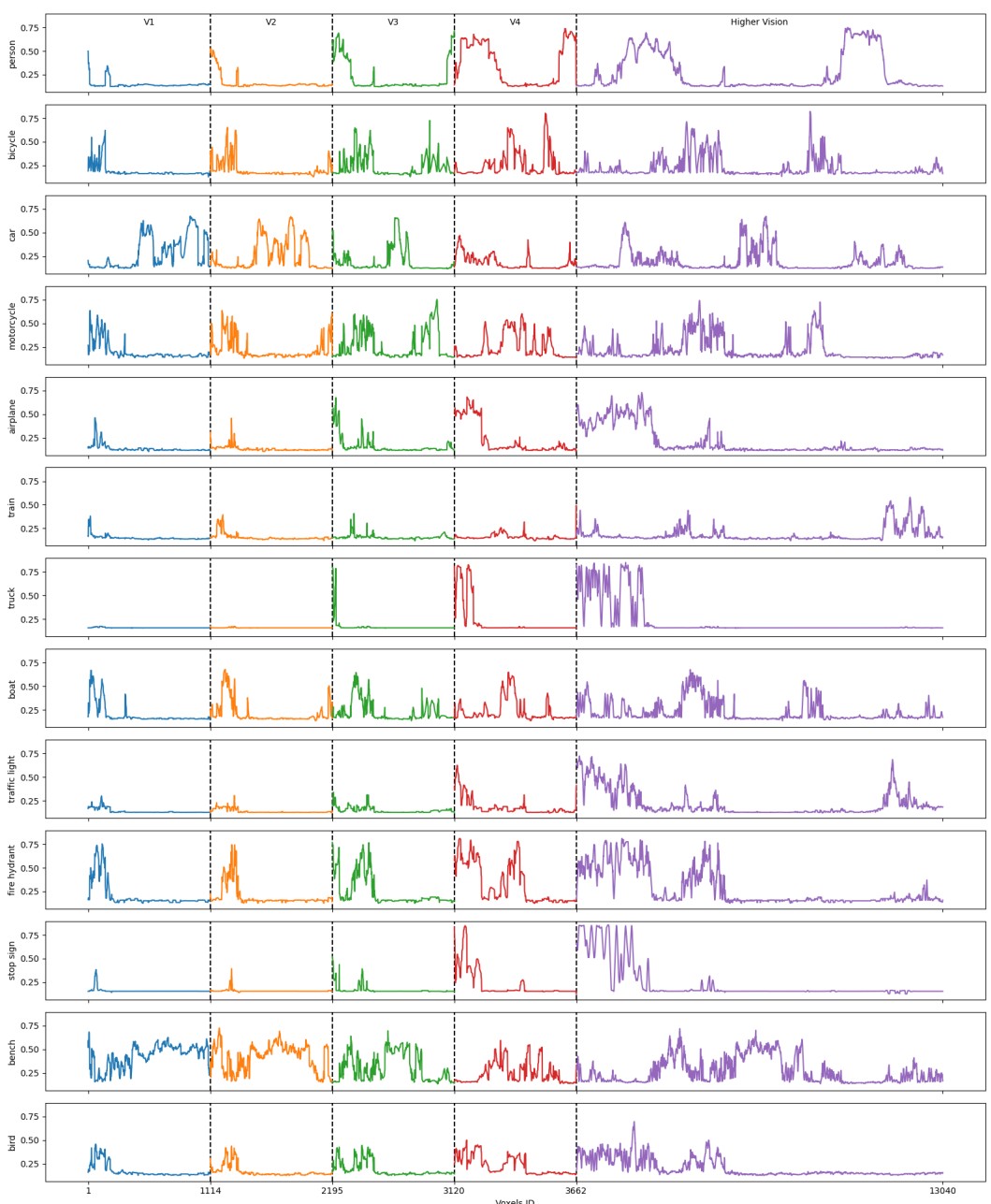

Figure 14: 1D Object-Voxel activations by brain vision ROIs for subj05.

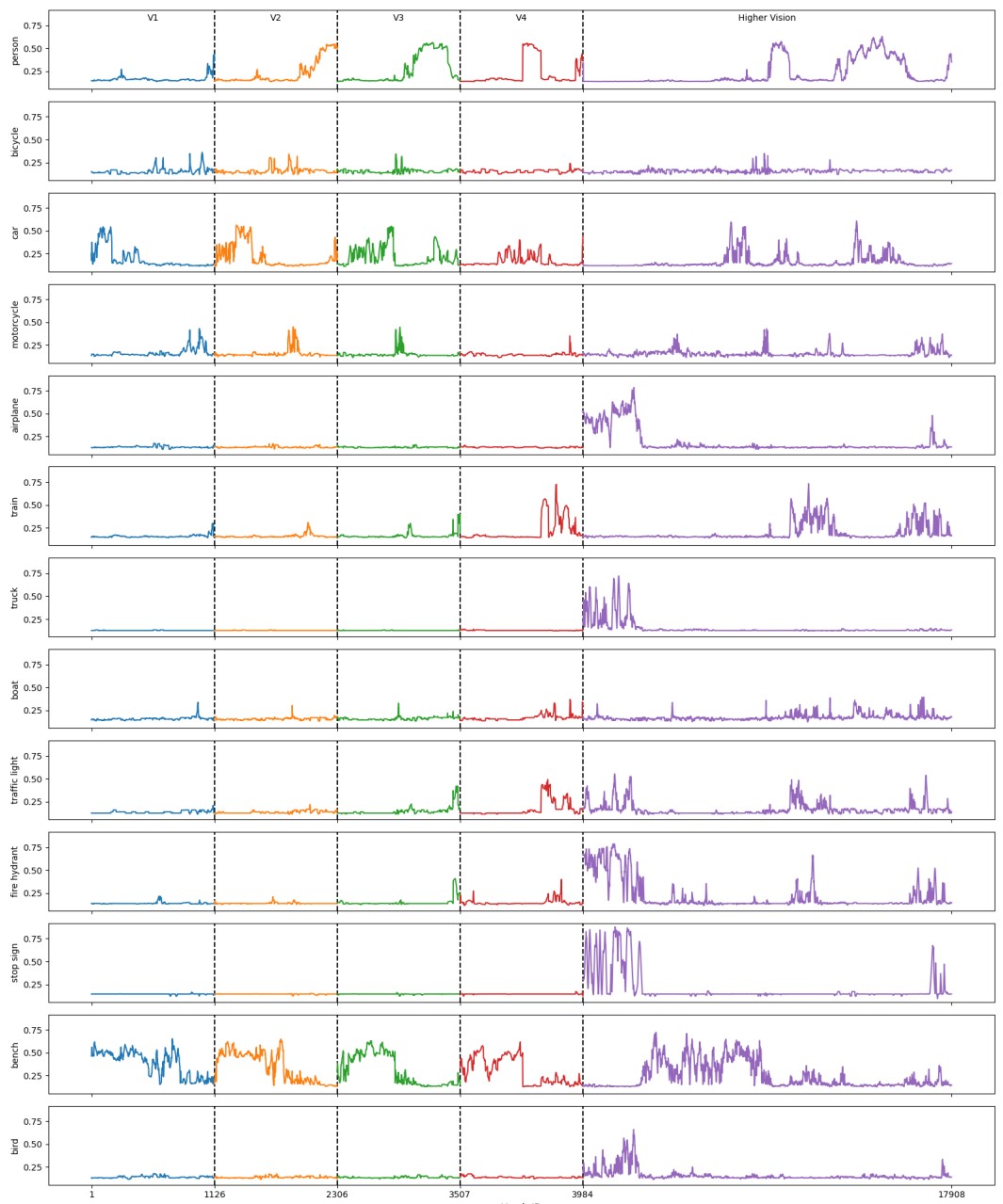

Figure 15: 1D Object-Voxel activations by brain vision ROIs for subj06.

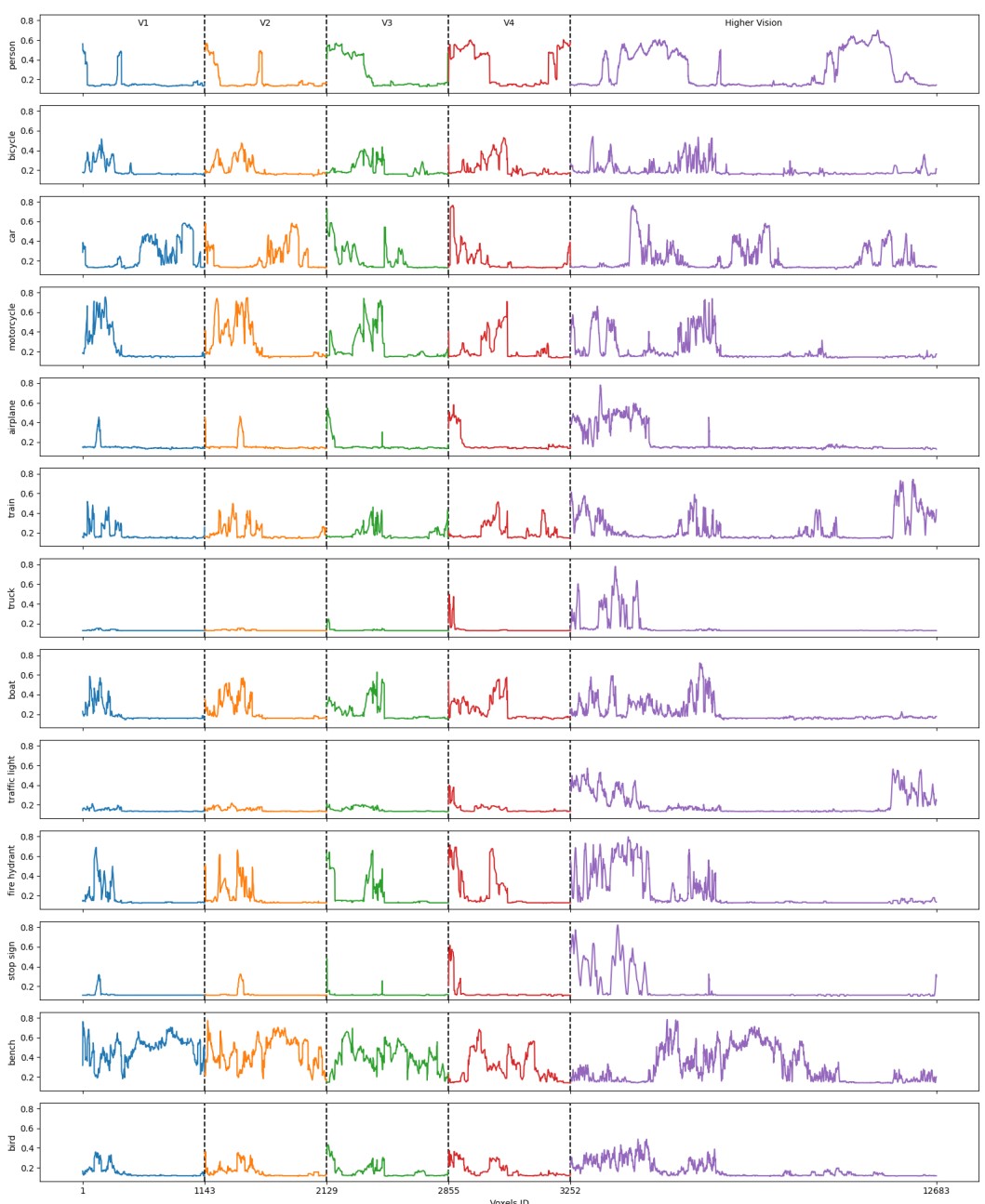

Figure 16: 1D Object-Voxel activations by brain vision ROIs for subj07.

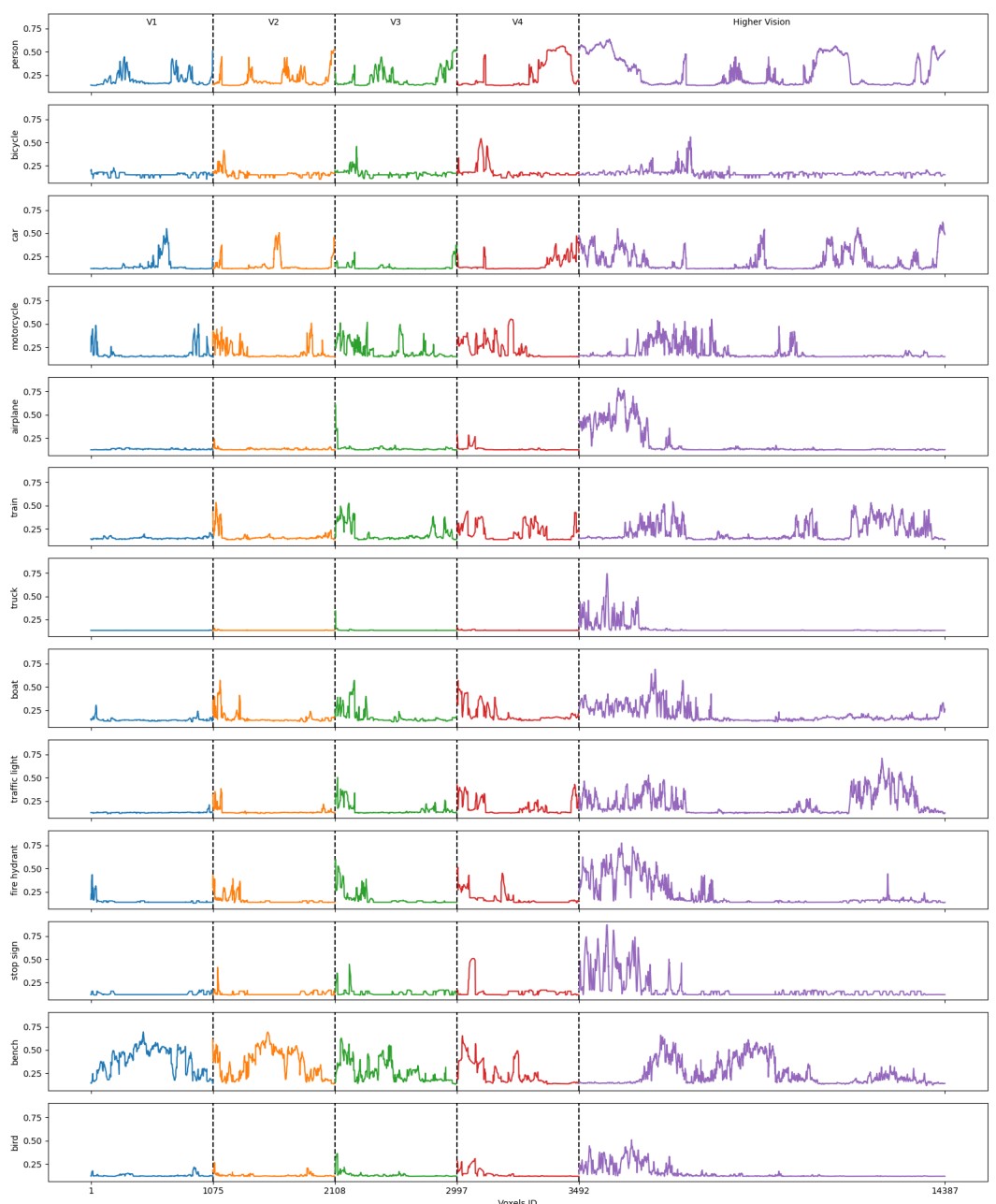

Figure 17: 1D Object-Voxel activations by brain vision ROIs for subj08.

### A.11.3 Cross-subject 3D Activation Pattern

Similar to Figure 4 in the main paper, we provide more visualization results on 3D Object-Voxel activation below:

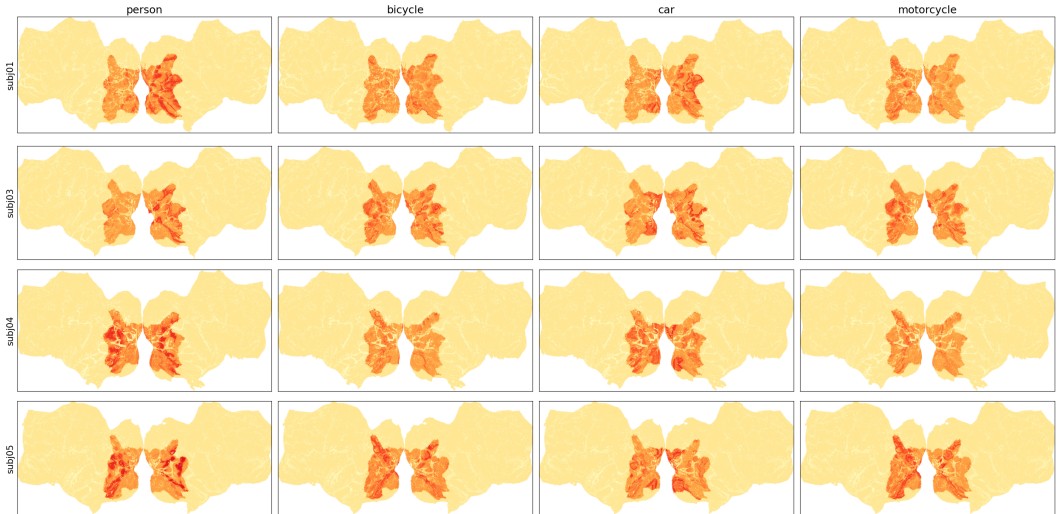

Figure 18: 3D Object-Voxel activations of *person*, *bicycle*, *car*, and *motorcycle* for subj01, subj03, subj04, and subj05.

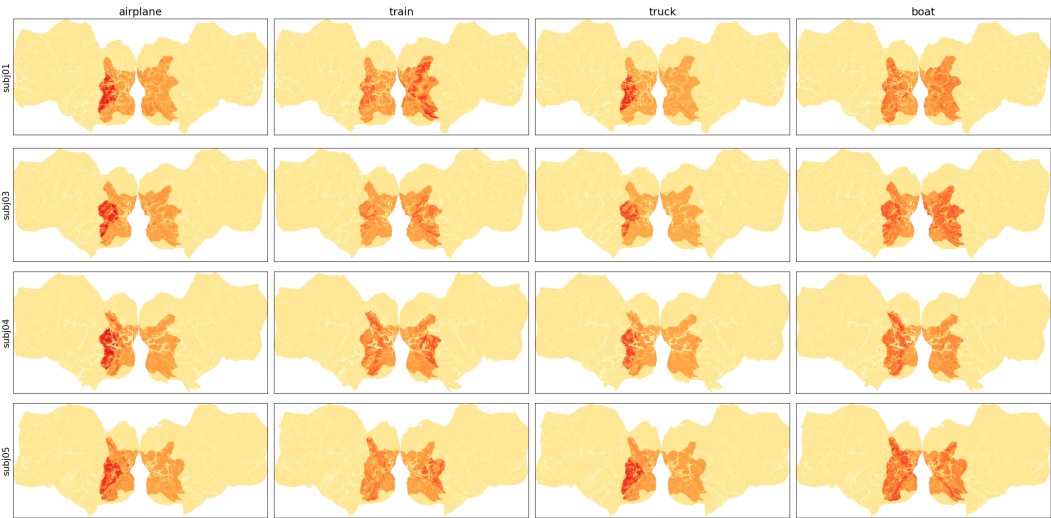

Figure 19: 3D Object-Voxel activations of *airplane*, *train*, *truck*, and *boat* for subj01, subj03, subj04, and subj05.

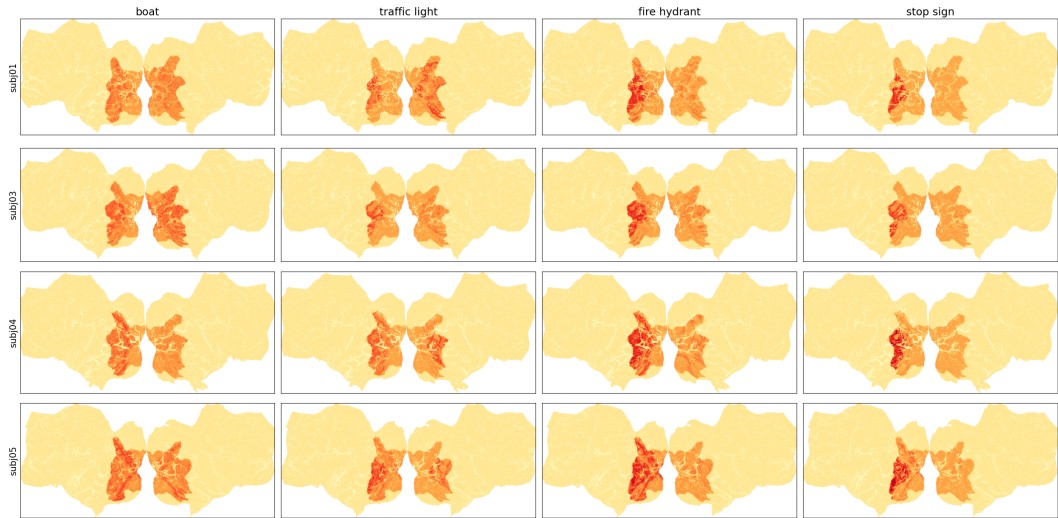

Figure 20: 3D Object-Voxel activations of *boat*, *traffic light*, *fire hydrant*, and *stop sign* for subj01, subj03, subj04, and subj05.

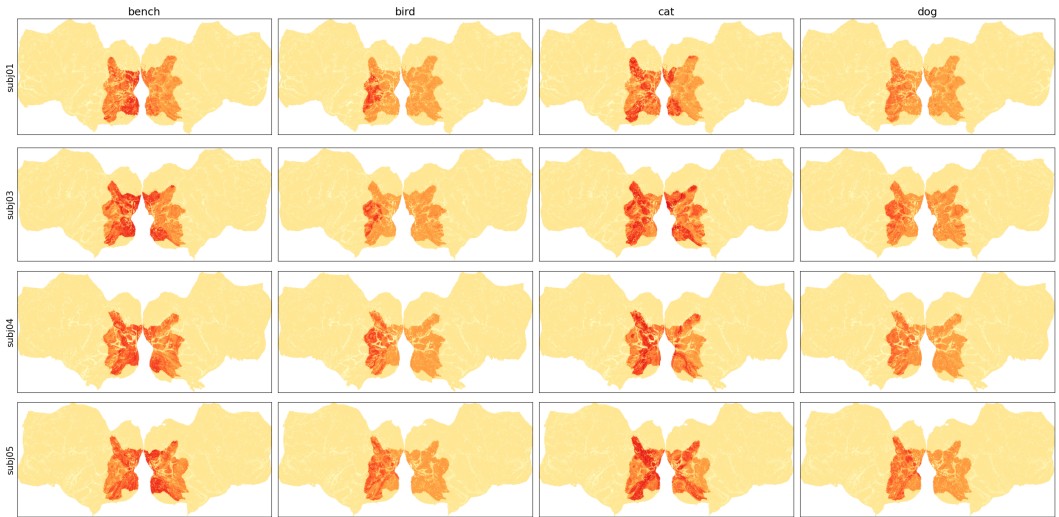

Figure 21: 3D Object-Voxel activations of *bench*, *bird*, *cat*, and *dog* for subj01, subj03, subj04, and subj05.

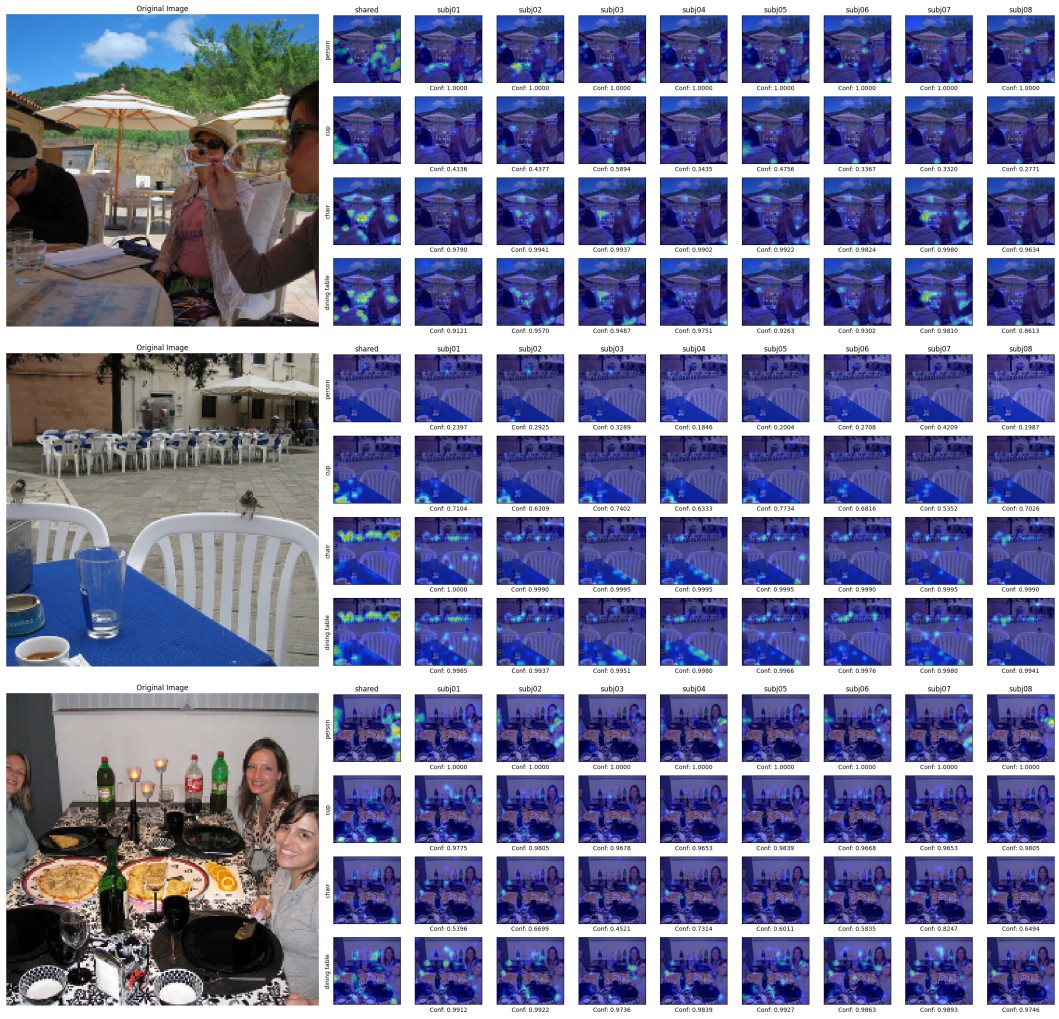

Figure 22: Variations in subjects' attention to different objects. Four objects: *person*, *cup*, *chair*, and *dining table* are selected for visualization.

### A.11.4 Variations in Subject Attention

Similar to Figure 4 in the main paper, we provide more visualization results on variations in subjects' attention to different objects in the same image. The leftmost image shows the visual stimulus. Plots in the second column represent the shared attention across all subjects, and the remaining eight columns show the residual, subject-specific attention alongside predicted probabilities to compare recognition confidence and priority.

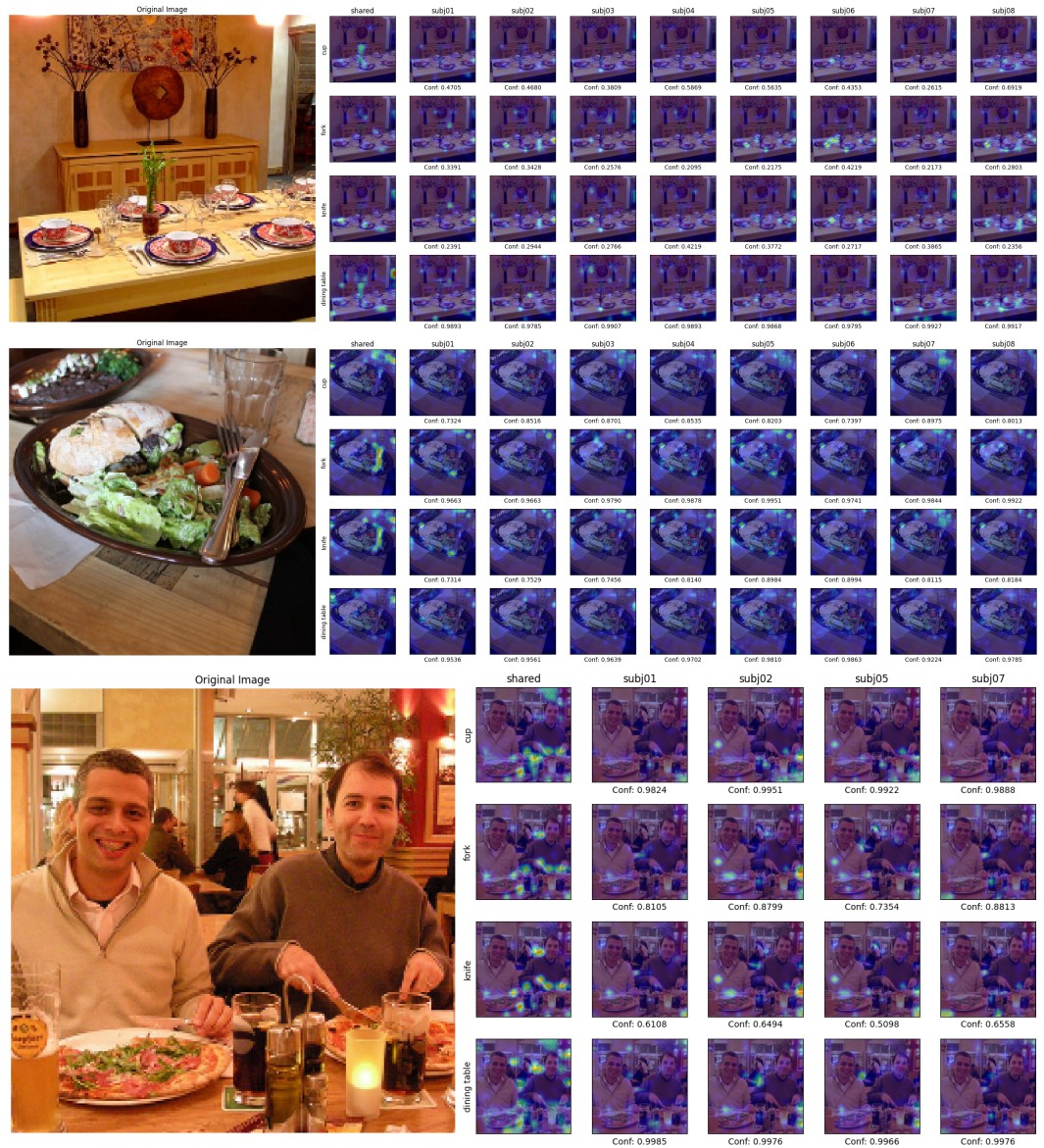

Figure 23: Variations in subjects' attention to different objects. Four objects: *cup*, *fork*, *knife*, and *dining table* are selected for visualization.

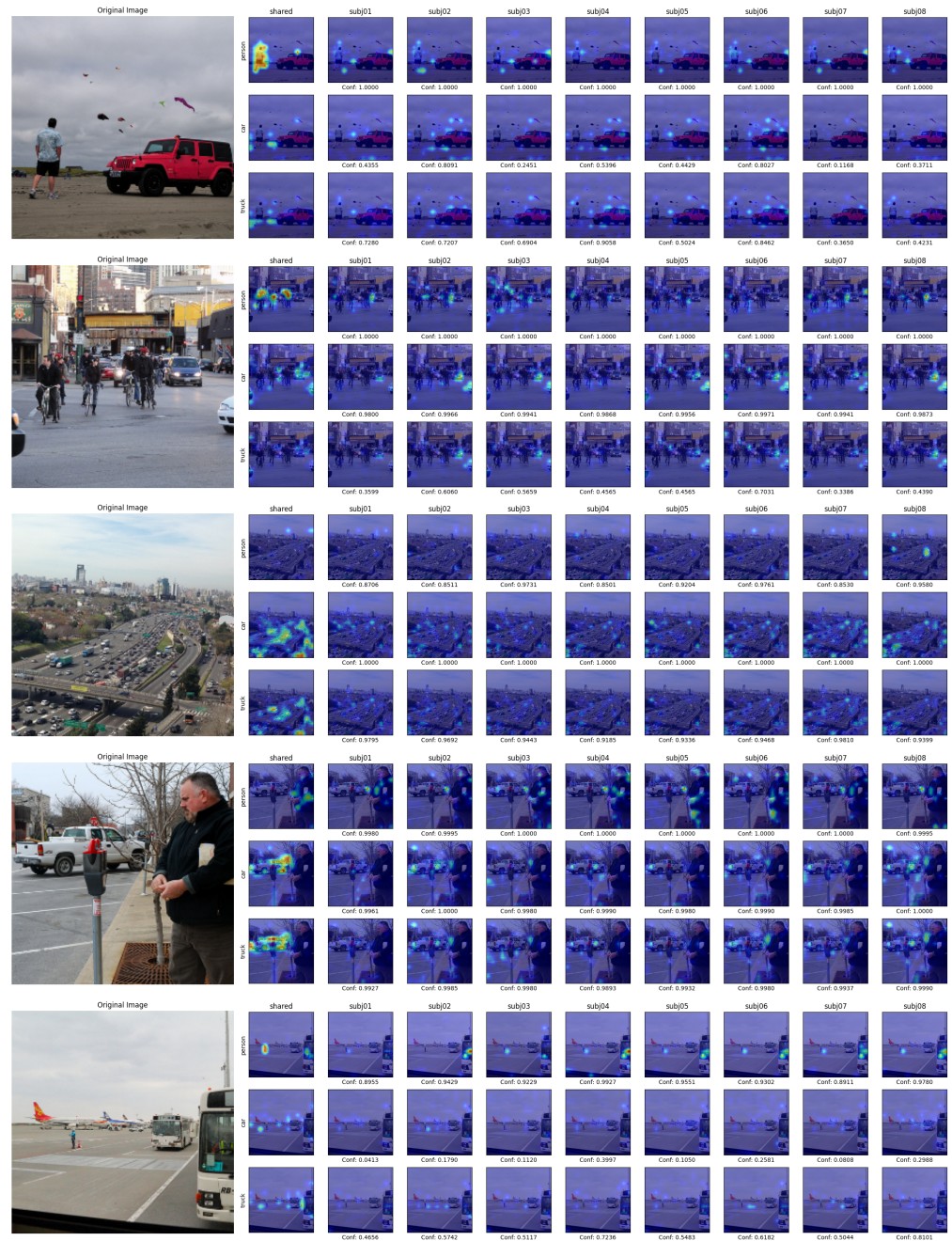

Figure 24: Variations in subjects' attention to different objects. Three objects: *person*, *car*, and *truck* are selected for visualization.

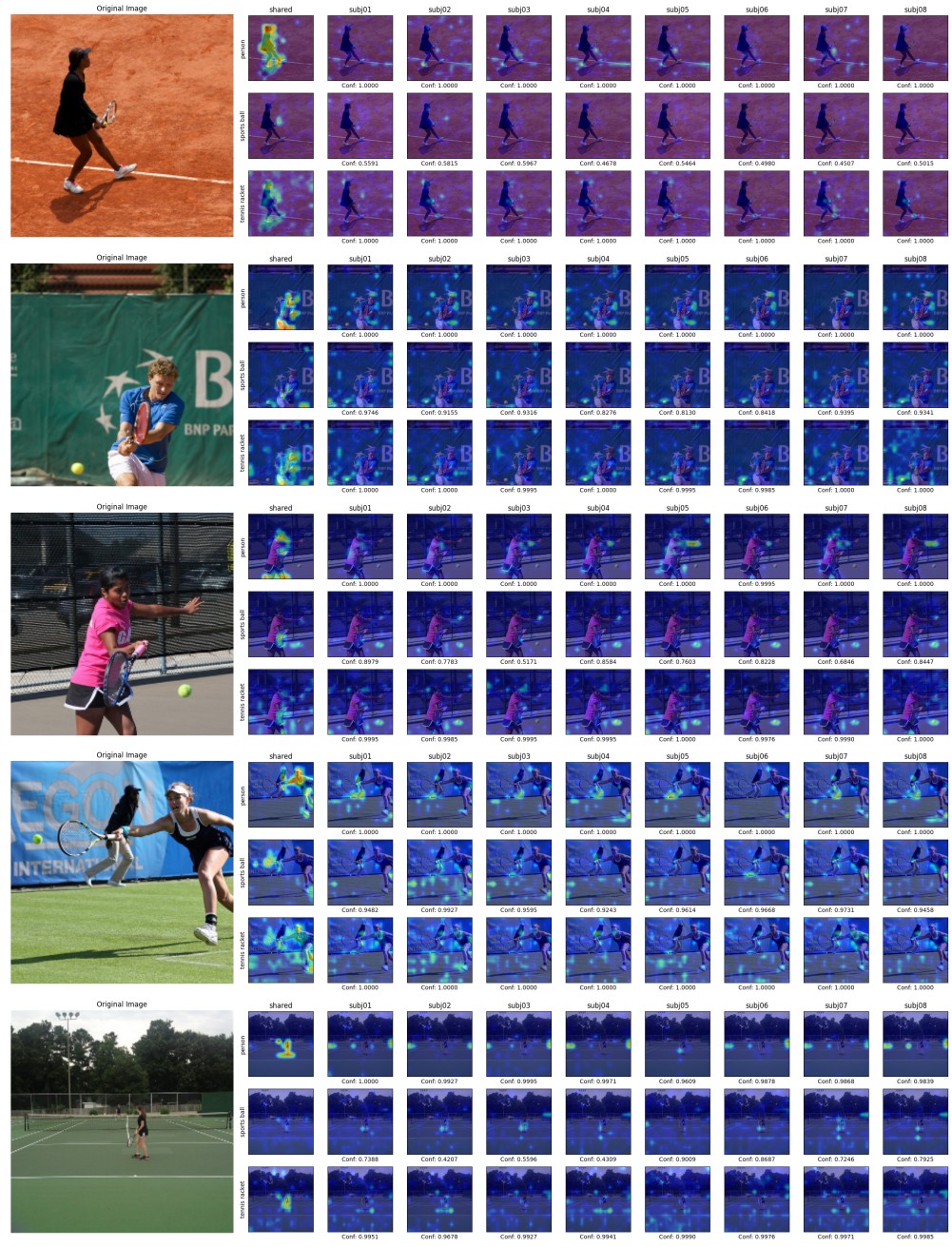

Figure 25: Variations in subjects' attention to different objects. Three objects: *person*, *sports ball*, and *tennis racket* are selected for visualization.

