# OpenReview forum: "$i$MIND: Insightful Multi-subject Invariant Neural Decoding"
_NeurIPS.cc/2025/Conference — NeurIPS 2025 poster_

### Official Review · Reviewer_1WBp · 2025-06-06

**Clarity:** 3
**Significance:** 3
**Originality:** 4
**Rating:** 5
**Confidence:** 5

**Summary:**

the authors propose iMIND, the insightful Multi-subject Invariant Neural Decoding model, that substantially advances the Neural Transcoding field. it is composed of a VIT based encoder which extracts features that will be decoupled into their object and subject-specific components to tackle the user identification and multi-label object classification.
results are promising, and the visualization of the attention overlay obtained from the model on the images is stunning.
a sufficient ablation study has been done, and the supplementary material completes the manuscript.

**Questions:**

- could you come up with a figure of the architecture?
- what are the pros of using a ViT-based encoder instead of other architectures?
- you did the subject identification task. why didn't you benchmark your model on this task on other fMRI datasets?
- in Tab. 1, what would the results have been if you also used text inputs, like in [50]?

**Ethical Concerns:**

["NO or VERY MINOR ethics concerns only"]

**Final Justification:**

After careful examination of the brilliant authors' rebuttal, my only concern for this manuscript remains the absence of a figure of the proposed architecture.
Overall, I confirm my previous rating and I am eager to recommend this paper for acceptance.

**Limitations:**

yes

**Quality:**

3

**Strengths And Weaknesses:**

S1 - the task is quite novel and interesting, and the paper is well-written
S2 - there are both quantitative (Tab. 1-2) and qualitative (Fig. 1-6) results shown, which seems promising and sound
S3 - there is a detailed ablation study
W1 - a figure of the proposed framework would have ease the flow and clarity of the entire manuscript
W2 - there is no code associated to the manuscript, hindering reproducibility (!!!)

---

> ### Author Rebuttal · Authors · 2025-07-30
>
> We sincerely appreciate Reviewer 1WBp for your questions and suggestions. Below are our responses. (W and Q stand for Weakness and Question, respectively.)
>
> # [W1 W2 Q1] Architecture Figure and Code
>
> Thanks very much for your great suggestion. At this moment, NeurIPS does not permit the inclusion of **rich media** or **external links** during the rebuttal phase this year. We **have prepared an architectural figure** and will include it in the final version to make the paper easy to follow. In addition, we would be happy to **share relevant portions of our code** in Markdown format during the discussion phase if you have concerns regarding **reproducibility**.
>
> # [Q2] ViT-based Encoder Justification
> Our choice of a ViT-based encoder for fMRI is motivated by both practical and representational considerations:
> - **Global Context Modeling**: Our fMRI inputs are flattened into **1D** vectors, which discards the original **spatial layout** of voxels. Unlike CNNs, which rely on **local receptive fields**, Vision Transformers (ViTs) have **global receptive fields**, enabling them to model **long-range dependencies across brain regions**. This helps **compensate** for the loss of **spatial structure** in our preprocessed inputs.
> - **Modality Alignment with CLIP**: Since the visual features we use are extracted from a pretrained **ViT-B/16 CLIP model**, using a ViT-based encoder for fMRI improves **architectural compatibility** and **facilitates cross-attention fusion**. This design choice reduces the **modality gap** and encourages more effective **alignment** between neural and visual representations.
>
> # [Q3] More Datasets
> We appreciate your comment on the generalization of the data set. We agree that demonstrating cross-dataset generalizability is an important step toward establishing a more comprehensive and robust framework.
>
> We introduce subject identification as an **auxiliary task** to support our main motivation: understanding the neural mechanisms of **visual processing in a data-driven manner**. This focus requires datasets that include paired natural visual stimuli and corresponding neural responses across multiple subjects. Given this requirement, only a few datasets are suitable—namely, NSD, HCP Movie, and BOLD5000.
>
> Among them, NSD is the only one that provides a **reproducible and well-documented benchmark** for **semantic object classification** tasks. In contrast, both **HCP Movie** and **BOLD5000** lack established benchmarks for classification-based decoding, and existing studies primarily focus on **reconstruction or resting-state analysis**. Moreover, models trained on these datasets often **do not release pre-trained weights**, and the corresponding **preprocessing pipelines** are either **unavailable or not well-documented**, making fair comparisons infeasible.
>
> We recognize the value of cross-dataset evaluation and plan to extend our experiments to HCP and BOLD5000 in future work. We will include this clarification in the final version of the manuscript.
>
> # [Q4] Text Modality
> Thank you for raising this point. We chose not to include the text modality in our current model for both practical and methodological reasons.
> - **Motivational Misalignment**: Our primary goal is to study semantic decoding from **vision-related brain activity**, using only **visual cortex voxels**. In the inference stage, it is not practical to assume text data **is available** for test samples. Introducing text features **risks semantic leakage**, as object names often appear explicitly in captions (e.g., in COCO descriptions), which could bias the model to **cheat on text** rather than decoding semantic content from fMRI. This would compromise our central objective of probing the brain's visual decoding mechanisms.
>
> - **Implementation Complexity**: The CLIP visual features used in our model are extracted from the image encoder **prior to** text-image alignment, meaning they **do not share a common latent space** with CLIP text embeddings. To incorporate text features, we would need to retrieve aligned image-text pairs and **re-extract features from the jointly aligned CLIP embedding space**. Additionally, the visual embeddings **before and after alignment differ in dimensionality**, which introduces **incompatibility** with our current architecture. It is also important to note that CLIP aligns **global image and text embeddings**, whereas our model operates on **local visual features for fine-grained semantic decoding**. Incorporating text would therefore require **substantial** changes to our input representation and necessitate retraining the entire model—including the cross-attention module and downstream components—along with extensive hyperparameter tuning. Due to time constraints during the review period, we were unable to carry out this significant modification.
>
> However, we consider your suggestions to be a **promising direction** for investigating **brain-language functionalities** in which we are very intereste and will certainly explore them in our future work.

---

> > ### Comment · Reviewer_1WBp · 2025-08-04
> >
> > Thank you for the clarifications.
> >
> > 1. I understand that you could not include images in the rebuttal, but the image of the architecture of your method should have been included in the main manuscript from the beginning.
> > 2. Thank you for the clarification, it seems solid
> > 3. and 4. I consider your current experiments sufficient for this submission, and hope to have inspired you for the future

---

> > > ### Author Response · Authors · 2025-08-08
> > >
> > > Thank you for your constructive feedback and for recognizing the clarifications in our rebuttal. We sincerely appreciate your time and valuable insights, which have helped strengthen our manuscript. We fully agree with your suggestion and commit to including the architecture figure in the revised manuscript. We are also grateful for your positive assessment of our experimental work and your encouraging remarks about future directions.

---

### Official Review · Reviewer_6F63 · 2025-06-25

**Clarity:** 4
**Significance:** 3
**Originality:** 3
**Rating:** 4
**Confidence:** 4

**Summary:**

Neural coding is a long standing question. While recent generative models provided a powerful tool for image reconstruction, it's performance largely depend on the generative model rather than the interpretability of neural activity. Furthermore, it is unclear how those models can handle subject-wise variability. While most work has been focused on minimizing inter-subject variability for generalizability , limited work aims to explicitly partition subject-specific differences and shared task-relevant features.

This work tried to disentangle subject-specific and stimulus/object-specific information by carefully designing a two stage training model and orthogonalization-constrained loss to identify corresponding subspaces in learned space of neural features. The first stage of the model is a ViT-based encoding model to identify the latent representation space of fMRI voxels in response to images. The second stage of training is dual-decoding stage with orthogonal constrains of the latent features. They achieved better performance in semantic decoding and biometric decoding tasks compared with some other models. As a standard control, model explanation and ablation analysis have been provided.

Key contributions are:
1. Modeling: a new dual-step encoding-decoding framework for subject-specific and object-specific components disentanglement.
2. Analysis: analysis of difference and variability across subjects.

**Questions:**

**Opportunities for Improvement:**
- add appropriate regularization to baseline model for a fair biometric decoding performance comparison
- Assumption of subject and object specific information are linearly entangled may need more justification. Further explanation and discussion of linear entanglement hypothesis.
- line 2-9: Description of gap and contribution is ambiguous. The method or idea could potentially offer more understanding but results of the paper are still limited in interpreting visual functionalities.
- $d_{obj}$ is treated as a user-defined value (line 146) but the interpretation of hyperparameter influence is weak (A.6). Further analysis of the subspace should be investigated from perspective of representation, dimensionality and dynamics.
- Biometric decoding is measured through subject classification. Baseline model may need more regularization such as sparsity to prevent overfitting due to high-dimensional input. (3.2)
- line 352: No evidence to show that the basis transformation is robust.

**Ethical Concerns:**

["NO or VERY MINOR ethics concerns only"]

**Final Justification:**

Based on the rebuttal and discussion. I would like to keep my original rating: 4.

**Limitations:**

Yes.

**Paper Formatting Concerns:**

No.

**Quality:**

3

**Strengths And Weaknesses:**

**Strengths:**:
The strengths of this paper includes 1. a new decoding framework for subject and semantic classification; 2. visualization of object-voxel activation patterns; 3. analysis of difference and variability across subjects.

- Quality: The effort to partition subject-specific vs object specific latents is of interest to the general community. The concern raised about reconstruction method for neural coding deserve more discussion as well.
- Clarity: The paper is clearly written and well organized, including its appendix. Loss function, model training, evaluation metric are clearly explained as well.
- Originality: This paper proposed a new model to disentangle subject-specific and stimulus/task-relevant information through data-driven multi-subject models.
- Significance: The question of subject-wise variability has been a big hurdle in field of neural coding and brain-machine interface. Finding a way to disentangle subject-specific variance from stimulus-specific variance is of great importance. Previous work like the Fisher information, Factor analysis and latent analysis have this flavor, but need to make many assumptions. This method suggested by the paper could be a new way to solve this problem.

*Weaknesses**:
- The concern that reconstruction method heavily depend on the generative model and cannot fully reveal neural coding properties is valid. However, directly reduce the dimensionality to object level or semantic level representation in fMRI does not directly address the question. There are limitations for models that can only perform object classifications as well. It is unclear what kind of features are captured for object discrimination.

- Subject and object classification loss is task-specific and may reveal only partial variance from individual-subject and stimulus. Once the latent space is disentangled, linear decoding performance could be high but the variance may not be complete. In other words, there are other variance, which could be pushed to null-space in the latent that got discarded by this method.

---

> ### Author Rebuttal · Authors · 2025-07-31
>
> We thank Reviewer 6F63 for your constructive feedback. Below are our responses. (W and Q stand for Weakness and Question, respectively.)
>
> # [W1] Semantic Decoding and Feature-level Interpretability
> We appreciate your insightful comments on the limitations of existing approaches and the challenges of interpretability in object-level neural decoding.
>
> Many recent works in neural decoding have **prioritized generating high-fidelity** visual reconstructions from fMRI data. While these outputs are visually impressive, such methods often rely heavily on pretrained generative models and focus primarily on image realism rather than understanding the **neural signals** themselves. As a result, they contribute **marginally** to advancing our knowledge of how the **brain encodes visual information**. This fundamental gap motivates our work: rather than asking **what the image looks like**, we ask **how semantic object representations are encoded and how they vary across individuals at the neural level**.
>
> In our work, we take **a discriminative decoding approach** that simultaneously predicts object categories (semantic decoding) and subject identities (biometric decoding) from fMRI signals. We agree that object classification alone **does not** fully capture the richness of neural representations. However, we argue that our framework provides **a meaningful step** toward interpretability by enabling **structured**, **task-relevant decomposition** of neural information.
>
> Our strong performance in semantic decoding allows us to leverage **post-hoc interpretability tools** (i.e., GradCAM, Attention Rollout) to analyze both model behavior and brain function. Specifically:
> - **Voxel-object activation maps (Fig. 2–3)** and **subject-specific attention analyses (Fig. 4–6)** uncover both shared and individualized patterns of neural activity in response to visual stimuli；
> - **These analyses reveal attention dynamics and voxel selectivity** that offer a data-driven perspective on how the brain represents object-level semantics—an aspect typically overlooked in reconstruction-based methods;
> - By combining **subject-object disentanglement** with **voxel-level interpretation**, we address a largely underexplored question: **how consistent are object-specific neural encodings across subjects, and where do individual differences emerge?**
>
> # [W2] Variance Pushed to the Null Space
> We appreciate this observation. It is indeed true that our disentanglement and decoding objectives **prioritize task-relevant variance**—specifically, for subject identification and object classification—which may lead to the **omission* of other latent dimensions not directly aligned with these tasks. This trade-off is inherent to discriminative frameworks that aim for linear separability in service of specific objectives.
>
> As clarified in Section 2.3 and Appendix A.2, our method is designed to **isolate** linearly separable components of neural representations relevant to subject and object decoding, rather than **capturing the entire variance** present in the fMRI signal. It is therefore expected that residual information—such as low-variance components or task-irrelevant patterns—may **reside in the orthogonal null space** of the learned latent basis.
>
> We view this as a **valuable opportunity** rather than a limitation. In fact, one of the longer-term goals of our framework is to first isolate task-relevant components in a structured, interpretable manner, and then **iteratively explore** the null space to investigate what information remains. This could include:
> - Unsupervised or contrastive objectives to identify structure in residual variance;
> - Temporal dynamics or contextual dependencies not captured by static object categories;
> - Subtle cognitive or emotional correlates that are not labeled but may still be encoded in the data;
> -Or even noise patterns that correlate with session-specific or scanner-specific artifacts.
>
> We believe this two-stage process—(1) identifying interpretable, task-relevant features and (2) probing the remaining space—is a promising direction toward more **transparent, comprehensive, and human-aligned** neural decoding models, and we plan to pursue this in future work.
>
> Regarding your concern about **transparency at the latent feature level**: we strongly agree that interpretability of individual latent dimensions remains **a fundamental challenge in deep learning** regardless of tasks or modalities. Our method **does not claim to assign direct semantic meaning** to each latent unit. Instead, we probe the structure of the latent space through aggregate behavioral and voxel-wise analyses. This work provides a framework that moves toward more interpretable models, and we view detailed latent-space analysis (e.g., basis axis meaning, feature attribution, spectral structure) as an important direction for future investigation.
>
> # [Q1 Q5] Additional Subject Baselines
> Thank you for the valuable suggestions regarding regularization for more robust and fair biometric decoding baselines.
>
> In response, we conducted additional experiments using MLP classifiers with varying levels of regularization and architectural complexity. As shown in the table below, we found that adding **L2 regularization and ReLU activation** significantly improves subject classification performance using raw fMRI voxel inputs. Notably, even a simple two-layer MLP with ReLU achieves near-perfect accuracy and MCC. We interpret this result as strong **empirical support** for our assumption that subject information in fMRI signals can be linearly disentangled from the joint latent space. We will elaborate on this point in our response to the **linearity assumption**.
>
> |**Layer(s)**|**Regulation**|**Activation**|**ACC**|**MCC**|
> |:--------------:|:----------------:|:----------------:|:-----------:|:-----------:|
> |1|—|—|.2833|.1812|
> ||L2|—|.3767|.2899|
> ||L2|ReLu|.5726|.5262|
> |2|L2|ReLu|**.9999**|**.9999**|
>
> # [Q2] Linear Assumption Justification
> We appreciate your concern regarding the justification of our linear entanglement hypothesis. Due to **space constraints** (10,000 characters), we were unable to include a detailed discussion in this response. We are truly sorry about this. However, **Reviewer T9by** raised the same point, and we have provided a thorough response there. We kindly refer you to our reply to **T9by** for further clarification on the detailed explanation and discussion of our linear entanglement hypothesis.
>
> # [Q4 Q6] Subspace Analysis and Robust Transformation
> We thank you for raising the insightful questions regarding the interpretation of the object-specific latent dimensionality $d_\text{obj}$ and its effect on the model’s representation and dynamics.
>
> ## Interpretation of $d_\text{obj}$ and Subspace Analysis
> While $d\_\text{obj}$ is treated as a user-defined hyperparameter in our formulation, we conducted **an additional analysis** to investigate its representational efficiency using **PCA** on the object-specific features $\mathbf{Z}\_\text{obj}$ with $d_\text{obj}=700$. We computed the **cumulative variance explained** and found that:
> - The first **10 components** explain approximately **92.27%** of the variance;
> - **30 components** capture over **98.05%**, and
> - Over **99.9%** of the total variance is explained by fewer than **120 dimensions**.
>
> Along with the performance stability shown in our ablation in Appendix A.6, this suggests that although we allocate 700 dimensions to $\mathbf{Z}\_\text{obj}$, the intrinsic dimensionality of the object-specific subspace is **significantly lower*. The fact that a small number of principal components **dominate** variance implies that the object subspace learned by our model is **highly structured, low-rank, and compressible**, indicating **meaningful latent representations** rather than arbitrary overparameterization. The model itself is **not sensitive** to the exact choice of $d_\text{obj}$ beyond a minimum threshold. These findings open up exciting potential for further exploration. In future work, we plan to study the **semantic interpretability** of leading PCA axes, examine temporal and **dynamic properties** of the object representations, and probe whether certain principal components correspond to high-level object categories or visual features.
>
> ## Robustness of Basis Transformation
> Regarding line 352, we agree that the term **robust** should be more precisely clarified. This robustness suggests that the disentangled space effectively captures semantic content across a range of dimensions in terms of dimensionality. We will revise the wording to **effective** or **consistent** if you feel robust is over-claimed here.
>
> # [Q3] Abstract Revision
> Thanks for pointing this out. We will revise our abstract to avoid ambiguity and include the following contents accordingly:
> While reconstruction tasks leverage powerful generative models to produce high-fidelity images from neural recordings, they often pay limited attention to the underlying neural representations and rely heavily on pretrained priors. As a result, they provide limited insight into which neural components encode semantic content or how these representations vary across individuals. Our work shifts the focus toward understanding the structure and variability of semantic encoding in the brain, rather than merely replicating visual appearance.
>
> In addition, we deeply appreciate your recognition of our work's potential to offer deeper insights into visual functionalities. To further strengthen the paper, we would be greatly thankful for a brief clarification on which specific aspects of neural visual processing could be more thoroughly illuminated. Your expert suggestions would be invaluable in guiding us to enhance the biological interpretability of our findings, whether in this revision or future work, while maintaining our core focus on discriminative decoding.

---

> > ### Comment · Reviewer_6F63 · 2025-08-04
> >
> > Thanks for making corresponding changes in the revision.

---

> > > ### Author Response · Authors · 2025-08-08
> > >
> > > Thank you for your acknowledgment of our revisions. We appreciate the time you have taken to review our manuscript and are grateful for your constructive feedback throughout the review process. Please don't hesitate to let us know if you have any additional comments or want to discuss further.

---

### Official Review · Reviewer_T9by · 2025-07-02

**Clarity:** 4
**Significance:** 3
**Originality:** 3
**Rating:** 5
**Confidence:** 4

**Summary:**

This paper introduces iMIND, a framework that aims to address the core challenge of visual brain decoding (using fMRI), particularly how to handle significant neural variability across individuals.

The framework's central idea is to explicitly disentangle the "subject-specific" and "object-specific" components of neural representations through a dual-decoding task (biometric and semantic decoding). Specifically, the model comprises three key steps:

1. A ViT-based masked autoencoder is used to establish a shared latent neural representation space across all subjects.

2. A learnable orthonormal basis transformation method disentangles the features in this shared space into two orthogonal subspaces: a "subject-specific" one and an "object-specific" one.

3. These disentangled features are then utilized for biometric identification (identifying the subject) and semantic classification (identifying the object the subject perceived).

Experiments demonstrate that iMIND achieves SOTA performance on the semantic decoding task and near-perfect accuracy on the biometric identification task. More importantly, the paper provides extensive analysis showing that the model can generate neuroscientifically meaningful "voxel-object activation maps." It also uncovers differences in attentional patterns among subjects viewing the same image or category, offering data-driven insights into the mechanisms of the brain's visual processing.

**Questions:**

1. Please see the points raised in the Weaknesses section above.

2. One more question: The authors state that Table 1 lists few methods due to the limited research on semantic neural decoding. However, other methods like MindBridge [1] could likely be adapted to create a baseline by substituting the task head for classification. It would be interesting to see how other methods would perform on this task.

[1] MindBridge: A Cross-Subject Brain Decoding Framework, CVPR 2024.

**Ethical Concerns:**

["NO or VERY MINOR ethics concerns only"]

**Final Justification:**

I am satisfied with the rebuttal and am happy to keep my original positive rating.

**Limitations:**

Yes

**Quality:**

4

**Strengths And Weaknesses:**

# Strength
1. Elegant and Effective Method Design: The idea of using a learnable orthonormal basis to decompose the feature space is elegant and technically sound, and the ablation study proves the effectiveness of this disentanglement. This "instructive strategy" is a significant conceptual advance over the prevailing "suppressive strategies" in the field.

2. Strong Experimental Results: The performance of iMIND on the semantic decoding task is superior to that of other methods. In the novel biometric decoding task, where no prior models exist for comparison, iMIND achieves near-perfect accuracy, providing powerful evidence for the effectiveness of its feature disentanglement framework.

3. In-depth and Insightful Analysis: The paper does not stop at reporting performance metrics; it dedicates significant effort to an interpretable analysis of the model, which is highly commendable. This includes:
> * By visualizing voxel-object activation maps, the paper successfully connects the model's learned representations with findings from neuroscience, such as the functional organization of the higher visual cortex and its sensitivity to specific categories like "person".
> * By examining prediction confidences and attention maps for objects like "chair," "cup," and "fork," the paper draws insightful conclusions about visual salience and cognitive strategies under temporal constraints.



# Weakness

1. The "Linear Entanglement" Assumption: The paper's core theory rests on a key assumption: that subject-specific and object-specific information are "linearly entangled" in the latent space. While the authors provide a justification (that deep networks aim to learn linearly separable features), this remains a strong assumption that may oversimplify the complex, non-linear dynamics of the brain. It is recommended that the authors discuss the limitations of this assumption more prominently in the main paper and speculate on how the model's performance might be affected if the assumption does not hold.

2. Large Performance Gap between fMRI+Image and fMRI-only Settings: Table 1 shows a massive performance gap between iMIND (fMRI-only) with an mAP of 0.310 and iMIND (fMRI+Image) with an mAP of 0.784. This large difference indicates that the image modality plays a decisive role. While the paper explains this is achieved via a cross-attention mechanism, it could be clearer about the precise role of the fMRI signal in this fusion. For instance, does the fMRI primarily provide a "where to look" (spatial attention) signal for the powerful CLIP features to then "identify"? Or is there a more complex synergy at play? Clarifying this would sharpen the paper's contribution.

---

> ### Author Rebuttal · Authors · 2025-07-31
>
> We sincerely appreciate Reviewer T9by for your valuable feedback. Below are our responses. (W and Q stand for Weakness and Question, respectively.)
> # [W1] Linear Entanglement Assumption
> We fully acknowledge that the brain’s functional dynamics are **fundamentally non-linear and complex**. Our assumption—that subject-specific and object-specific components are linearly entangled at the latent feature level—is **a simplifying inductive bias** introduced to enable interpretable, computationally tractable disentanglement. We **do not* claim it fully captures the richness of brain representations; rather, it serves as **a first-order approximation** that enables clear factorization of subject identity and semantic content from fMRI signals.
>
> ## Empirical Justification
> While we agree this assumption may not fully capture the brain's non-linear functional structure, we provide the following empirical evidence to support its pragmatic utility in our framework:
> - **Near-perfect biometric decoding (Table 2 in main paper)**: We achieve 99.9% accuracy in subject classification, indicating subject-specific features **can be linearly separated** from entangled fMRI features in the context of multi-subject learning;
> - **Subject-invariant object activations (Fig. 1 in main paper)**: After disentanglement, we observe that object activation patterns in the latent space exhibit **minimal cross-subject variability**, indicating that our method successfully distills **stable, shared semantic representations**—further supporting the effectiveness of linear disentanglement.
> - **Shallow MLP baselines confirm separability (Table below)**: We additionally implemented two-layer MLPs with ReLU activation for subject classification from raw fMRI. Surprisingly, even this **lightweight non-linear** model achieved near-perfect accuracy, validating the notion that subject-specific features are already largely linearly separable at the voxel level. This further justifies our design, especially considering our ViT encoder has **significantly higher representational capacity** compared with MLPs.
>
> | **Layer(s)** | **Regulation** | **Activation** | **ACC**   | **MCC**   |
> |--------------|----------------|----------------|-----------|-----------|
> | 1            | —             | —             | .2833    | .1812    |
> |             | L2             | —             | .3767    | .2899    |
> |             | L2             | ReLu           | .5726    | .5262    |
> | 2            | L2             | ReLu           | **.9999**| **.9999**|
>
> ## Discussion of Limitations
> We agree it is important to discuss potential implications if the linearity assumption **does not fully hold**:
> - If subject-object interactions are fundamentally **non-linear**, the disentangled object representation $\mathbf{Z}_\text{obj}$ may still **retain residual subject-specific** information, potentially introducing **subject bias** in semantic decoding and **diminishing our model's generalizability* for unseen subjects.
> - Conversely, enforcing strict linear disentanglement may **suppress relevant non-linear** object features in $\mathbf{Z}_\text{obj}$, potentially **smoothing out** sharp voxel-object modulations or **degrading decoding performance** for fine-grained categories.
>
> ## Future Directions
> We are actively exploring **non-linear disentanglement techniques**—e.g., manifold learning, kernel-based projection, or contrastive objectives—to model richer subject-object interactions while preserving **interpretability**. Incorporating auxiliary data such as **structural MRI** or **resting-state connectivity** could also regularize the subject basis more effectively. As suggested, we will make these limitations and future directions more explicit in the revised manuscript, particularly in the Discussion and Limitations sections.
>
> # [W2] Role of CLIP and fMRI
> We thank you for this thoughtful suggestion. Further clarifying the role of the fMRI signal in the cross-attention fusion is essential for us to highlight our contribution.
>
> To answer directly: in our fusion framework, the fMRI signal plays the role of **"what to attend to"** rather than simply "where to look." The fused output remains **fundamentally neural in nature**, modulated by subject-specific brain responses rather than being dominated by CLIP features. This is reflected in the design of the cross-attention module (Eq. 9), where **CLIP embeddings serve as queries**, while **fMRI features act as keys and values**—allowing neural responses to govern which semantic components are emphasized.
> This design achieves two complementary goals:
> - **Semantic Prioritization**: CLIP features encode **subject-invariant, stimulus-driven** semantic content and contain rich spatial and conceptual structure. During cross-attention, CLIP features **query the fMRI signal**, and the fMRI effectively **responds with what concepts or object components are prioritized**, based on subject-specific neural responses. In other words, attention weights reflect which parts of the CLIP space align with neural activations.
> - **Subject-Specific Modulation**: As shown in Fig. 4 (columns 3–10), residual attention maps reveal that fMRI signals modulate the CLIP-based predictions in a **subject-dependent** manner. This synergy allows our model to preserve semantic content from CLIP while capturing **individual-specific neural patterns**, highlighting how different subjects may focus on different semantic attributes of the same visual input.
>
> Thus, rather than acting as a passive filter or spatial mask, the fMRI features dynamically shape which aspects of the CLIP semantic space are elevated—creating a **bi-directional synergy**. CLIP provides **a semantically grounded reference frame**, while fMRI reveals how the **brain selectively engages** with that semantic structure.
>
> # [Q2] Comparison against MindBridge
> We thank you for this excellent suggestion and fully agree that including additional adapted baselines could strengthen the evaluation. In response, we adapted **MindBridge**—a reconstruction-oriented model—into a semantic decoding baseline by replacing its generative output head with a linear multi-label classification head, trained on its latent representations. While MindBridge was not originally designed for classification, this setup enables a fairer comparison of its representational utility under our semantic decoding task.
>
> Due to time and resource constraints, we followed MindBridge’s original training protocol and evaluated the model on **4 subjects (1, 2, 5, 7)**. In contrast, our model (iMIND) was trained and evaluated on **all 8 subjects**, using the same data split and evaluation metrics. The results are shown below:
> | **Methods** | **Modalities** | **mAP ↑** | **AUC ↑** | **Hamming ↓** |
> |--------------|----------------|-----------|-----------|----------------|
> | MindBridge   | fMRI+Image     | .1499| .8001| .0286     |
> |              | fMRI+Image+Text| .1521| .7999| .0289     |
> | iMind   | fMRI           | **.3095**| **.9132**| **.0271**     |
> |              | fMRI+Image     | **.7836**| **.9840**| **.0121**     |
>
> Despite the limited setup, these results demonstrate a **substantial performance gap**, especially in the **fused modality setting**. This gap reinforces a central argument of our paper: **semantic decoding—particularly in a multi-subject setting—requires models explicitly designed to handle subject variability and semantic disentanglement**. Simply reusing a reconstruction model, even with strong latent priors, is **insufficient**.
>
> We will include this analysis and baseline comparison in the final version to strengthen the experimental evaluation and better contextualize iMIND’s advantages.

---

> > ### Comment · Reviewer_T9by · 2025-08-07
> >
> > Thank you for the detailed rebuttal, which has addressed my concerns well. Looking forward to seeing these important clarifications reflected in the final manuscript. And I am happy to keep my original positive rating.

---

> > > ### Author Response · Authors · 2025-08-08
> > >
> > > Thank you for your kind feedback and for acknowledging our revisions. We sincerely appreciate your time and constructive comments, which have helped improve our manuscript. We commit to carefully incorporating all important clarifications into the final version as suggested, and we are grateful for your continued support.

---

> ### Author Response · Authors · 2025-08-05
>
> Thank you again for your time and valuable feedback on our paper. We wanted to kindly check if you have had a chance to review our rebuttal responses or if there are any points we could clarify further to assist your evaluation. We would be happy to continue the discussion.

---

### Official Review · Reviewer_7Y8L · 2025-07-03

**Clarity:** 2
**Significance:** 3
**Originality:** 3
**Rating:** 5
**Confidence:** 4

**Summary:**

This paper proposes iMIND, a dual-decoding framework for neural signal interpretation that disentangles fMRI data into subject-specific and object-specific components through orthogonal basis transformation. The method achieves state-of-the-art performance in both biometric identification and semantic classification tasks while providing interpretable voxel-object activation patterns and revealing individual differences in visual attention.

**Questions:**

1. The core claim of your method is that Zsubj and Zobj are properly disentangled, but no empirical validation is provided. Can you compute and report the linear correlation coefficients between these components? What is the mutual information between them?
2. Please clearly specify whether the voxel-object activation patterns and attention variation analyses (Sections 3.4-3.5) are based on the fMRI-only model or require the fMRI+Image multimodal version.
3. Can you include comparisons with VAE-based self-supervised fMRI encoders and other recent neural representation learning methods? The current baselines are insufficient for a comprehensive evaluation.

**Ethical Concerns:**

["NO or VERY MINOR ethics concerns only"]

**Final Justification:**

Thanks for the detailed response, I would keep the positive rating.

**Limitations:**

The authors should more thoroughly address several limitations: (1) evaluation is limited to a single dataset, (2) the lack of empirical validation for the claimed disentanglement, and (3) the dependency of interpretability analyses on multimodal inputs needs clarification.

**Paper Formatting Concerns:**

None.

**Quality:**

3

**Strengths And Weaknesses:**

Strengths
1. The orthogonal basis transformation approach for disentangling subject-specific and object-specific neural features is theoretically sound and well-motivated.
2. The paper demonstrates superior performance across multiple metrics (mAP, AUC, Hamming distance) and provides extensive visualizations of neural activation patterns, attention variations, and cross-subject consistency.
3. The voxel-object activation fingerprints and subject-specific attention pattern analysis provide valuable insights into individual differences in visual processing, which is rare in neural decoding literature.
4. Achieves state-of-the-art performance in semantic decoding (mAP: 0.310 vs 0.258 for previous best) and near-perfect biometric classification (ACC: 0.999).

Weaknesses
1.  The motivation connecting fMRI research to autonomous driving and medical diagnosis annotation problems is unconvincing and irrelevant. The paper should focus on neuroscience motivations rather than unrelated computer vision applications.
2. The paper lacks comparison with other self-supervised neural encoding methods, particularly VAE-based approaches for fMRI representation learning. The biometric decoding baselines are too simplistic.
3. Despite the theoretical completeness of the orthogonal decomposition, the paper fails to empirically validate that Z_subj and Z_obj are actually disentangled. Critical spillover tests are missing, such as computing linear correlation coefficients between the two components or measuring mutual information to verify independence.
4. It's ambiguous whether insightful neural activation and attention analyses depend on the fMRI-only model or require the fMRI+Image multimodal version.
5. Only evaluated on NSD dataset; generalization to other fMRI datasets remains unclear.

---

> ### Author Rebuttal · Authors · 2025-07-31
>
> We thank Reviewer 7Y8L for your constructive feedback. Below are our responses. (W, Q, and L stand for Weakness, Question, and Limitation, respectively.)
>
> # [W1] Motivation Rephrase
> Thanks very much for your great suggestion. We admit that the original introduction placed too much emphasis on downstream applications such as autonomous driving and medical diagnosis, which may **distract** from the core neuroscience motivations of our work. In the final version, we will rephrase the introduction to better highlight key neuroscience-driven tasks—such as **how the human brain encodes, disentangles, and generalizes semantic representations across individuals**—and to emphasize the role of fMRI as a powerful non-invasive tool for probing the visual processing pipeline and subject-specific perception mechanisms. We will highlight our goal of contributing to a deeper understanding of **multi-subject neural decoding** and **inter-individual variability** in visual perception. We will retain only one or two sentences (in the conclusion) about potential long-term implications for human-AI interaction, as suggested by recent NeurIPS trends in brain-inspired AI.
>
> # [W3 Q1 L2] Empirical Validation of Disentanglement
> This is a great point to make our work complete. To rigorously validate disentanglement, we have included the following results:
> - **Near-Zero Linear Correlation**: We calculate the correlation of each feature vector of $\mathbf{Z}\_\text{obj}$ and $\mathbf{Z}_\text{subj}$ and report their mean and standard as $0.0027\pm0.0012$, supporting our disentanglement claim;
> - **Orthogonality Enforcement**: The Frobenius norm $||\mathbf{B}\_\text{obj}\mathbf{B}\_\text{subj}^\top||= 2.7997\times 10^{-5}$ along with our orthonormal loss $\mathcal{L}_\text{orth}=7.2874\times10^{-5}$ confirms the role of learned $\mathbf{B}$ as an orthonormal basis for object-subject disentanglement.
>
> # [W2 Q3] Baseline Comparisons
> We have expanded our comparisons to include state-of-the-art self-supervised fMRI encoders and strengthened our biometric decoding baselines. Below, we summarize the key results:
> - For **semantic decoding** and **neuro encoding** methods, we evaluated **three more self-supervised** fMRI encoders, all trained with identical data/hyperparameters as shown below. We found that the **Masked Autoencoder (MAE)**, which serves as the backbone of our model, outperforms the other three self-supervised methods. This is primarily due to its ability to **reduce noise and redundancy inherent in fMRI data**, as also observed in Mind-Vis. Furthermore, since prior work on semantic decoding—particularly for **both classification and multi-subject modeling**—is limited, we include MindBridge for comparison, even though it was originally designed for reconstruction tasks. To adapt MindBridge for classification, we extract its output features and train a linear classification head on top. This comparison confirms the need for **task-specific architectures like ours that are explicitly designed for semantic decoding across subjects**.
>
> | **Backbone** | **Modalities** | **mAP ↑** | **AUC ↑** | **Hamming ↓** |
> |:--------------:|----------------|-----------|-----------|:----------------:|
> |MindBridge| fMRI+Image|.1499|.8001|.0286|
> ||fMRI+Image+Text|.1521|.7999|.0289|
> |AE|fMRI|.1474|.8042|.0377|
> ||fMRI+Image|.7051|.9725|.0169|
> |DAE|fMRI|.1435|.8012|.0379|
> ||fMRI+Image|.7135|.9748|.0159|
> |VAE|fMRI|.1290|.7230|.0374|
> ||fMRI+Image|.6951|.9683|.0165|
> |MAE (Ours)|fMRI|**.3095**|**.9132**|**.0271**|
> ||fMRI+Image|**.7836**|**.9840**|**.0121**|
>
> - For **biometric decoding**, we included **three** additional baseline models. Notably, we found that even a simple **two-layer MLP with ReLU activation and L2 regulation** can achieve perfect subject classification from fMRI voxel signals. This result provides strong empirical support for the core assumption of iMIND—that subject information in fMRI can be linearly separable at the latent feature level. Given that the Vision Transformer (ViT) offers **significantly greater feature extraction capacity** than a shallow MLP, this finding reinforces the validity of our disentanglement approach.
>
> |**Layer(s)**|**Regulation**|**Activation**|**ACC**|**MCC**|
> |:--------------:|:----------------:|:----------------:|:-----------:|:-----------:|
> |1|—|—|.2833|.1812|
> ||L2|—|.3767|.2899|
> ||L2|ReLu|.5726|.5262|
> |2|L2|ReLu|**.9999**|**.9999**|
>
> # [W5 L1] Dataset Generalization
> We appreciate your comment on the generalization of the data set. We agree that demonstrating cross-dataset generalizability is an important step toward establishing a more comprehensive and robust framework.
>
> In exploring additional datasets, we considered both the BOLD5000 and HCP movie-watching datasets, which involve naturalistic visual stimuli and multiple subjects. However, two key challenges currently limit their use in our study:
> - **Lack of semantic decoding benchmarks (BOLD5000)**: Unlike NSD, which provides well-curated annotations and is widely used for semantic decoding tasks, BOLD5000 lacks **standardized, reproducible benchmarks** for object-level classification. This makes it difficult to **evaluate and compare performance** under our semantic decoding setup.
> - **Computational challenges (HCP)**: The HCP movie dataset includes **158** subjects and more than **800** semantic concepts. Given the need to model multi-subject disentanglement and high-dimensional voxel features, applying our method to this dataset is **computationally demanding**. Unfortunately, due to limited time and resources during the review period, we were unable to complete a full evaluation.
>
> That means, we recognize the importance of cross-dataset validation and plan to explore generalization on datasets like HCP and BOLD5000 in future work. We will explicitly acknowledge this limitation in the revised manuscript and elaborate on it in the Limitations and Future Work section.

---

> ### Author Response · Authors · 2025-08-05
>
> Thank you for taking the time to review our rebuttal. If there are any points where we could further improve clarity or address remaining questions, please don’t hesitate to let us know. We would be happy to continue the discussion. Thank you again for your thoughtful feedback.

---

### Note · Authors · 2025-08-15

We thank the area chair and the reviewers for your constructive feedback and engaging discussion, which have helped strengthen our work. Below, we summarize the strengths recognized by reviewers, the main concerns raised, and how our rebuttal addressed them.
## Strengths
Reviewers highlighted several notable contributions and merits of our work:
- Novel and Elegant Methodology: A theoretically solid orthogonal basis transformation that disentangles subject-specific and object-specific neural features, representing a conceptual advance in addressing the persistent challenge of subject-wise variability in neural decoding research.
- SOTA Performance: Superior semantic decoding with consistent gains across three metrics and near-perfect biometric classification accuracy.
- Interpretability and Insightful Analyses: Comprehensive interpretability studies, including voxel–object activation fingerprints, attention variation analysis, and cross-subject consistency. These connect model behavior to neuroscience findings, providing rare insights into individual differences in visual processing.
## Concerns Addressed
We carefully considered all critical questions and, in our rebuttal, provided additional analyses and clarifications:
- Incorporated variants of self-supervised fMRI backbones and additional fMRI-learning models for a more complete semantic decoding comparison.
- Extended biometric decoding baselines, further supporting our linear entanglement assumption.
- Introduced three metrics to empirically confirm effective disentanglement of subject and object features.
- Elaborated on the roles of CLIP and fMRI features in our design, explaining their alignment with our motivation and objectives.
- Committed to wording refinements for the final version, including clearer introductory sentences, a more explicit abstract, expanded discussion of design choices, and the addition of an architectural figure to improve clarity.

Reviewers responded positively to the updates and noted that their concerns were fully addressed.
## Conclusion
We believe our work offers both methodological and scientific advances: a novel and effective framework for multi-subject invariant neural decoding, supported by strong empirical results and neuroscience-aligned analyses. Its ability to jointly achieve SOTA semantic & biometric decoding—while revealing activation and attention patterns—makes it a valuable contribution to both the machine learning and cognitive neuroscience communities.

---

### Decision · Program_Chairs · 2025-09-17

**Decision:**

Accept (poster)

**Comment:**

The paper introduces iMIND, a dual-decoding framework that disentangles subject-specific and object-specific fMRI representations for both biometric and semantic neural decoding. The approach is theoretically well-grounded, strongly motivated, and supported by extensive experiments with state-of-the-art results and insightful neuroscientific analyses. All reviewers were positive and found the rebuttal thorough in addressing their concerns.
**The main limitation is the lack of broader comparisons to more recent baselines, which somewhat narrows the empirical context,** while this tempers the strength of the contribution, the conceptual novelty and demonstrated effectiveness still justify acceptance.